# Single-cell full-length total RNA sequencing uncovers dynamics of recursive splicing and enhancer RNAs

Tetsutaro Hayashi [1], Haruka Ozaki [1], Yohei Sasagawa [1], Mana Umeda[1], Hiroki Danno [1] & Itoshi Nikaido [1,2]

Total RNA sequencing has been used to reveal poly(A) and non-poly(A) RNA expression, RNA processing and enhancer activity. To date, no method for full-length total RNA sequencing of single cells has been developed despite the potential of this technology for single-cell biology. Here we describe random displacement amplification sequencing (RamDA-seq), the first full-length total RNA-sequencing method for single cells. Compared with other methods, RamDA-seq shows high sensitivity to non-poly(A) RNA and near-complete full-length transcript coverage. Using RamDA-seq with differentiation time course samples of mouse embryonic stem cells, we reveal hundreds of dynamically regulated non-poly(A) transcripts, including histone transcripts and long noncoding RNA *Neat1*. Moreover, RamDA-seq profiles recursive splicing in >300-kb introns. RamDA-seq also detects enhancer RNAs and their cell type-specific activity in single cells. Taken together, we demonstrate that RamDA-seq could help investigate the dynamics of gene expression, RNA-processing events and transcriptional regulation in single cells.

---

[1] Bioinformatics Research Unit, Advanced Center for Computing and Communication, RIKEN, 2-1 Hirosawa Wako, Saitama, 351-0198, Japan. [2] Single-cell Omics Research Unit, Center for RIKEN Center for Developmental Biology, RIKEN, 2-1 Hirosawa Wako, Saitama, 351-0198, Japan. Tetsutaro Hayashi and Haruka Ozaki contributed equally to this work. Correspondence and requests for materials should be addressed to I.N. (email: itoshi.nikaido@riken.jp)

Total RNA sequencing (total RNA-seq) provides rich information on biological systems in addition to the abundance of mRNAs. First, total RNA-seq can measure not only poly(A) but also non-poly(A) RNAs, including nascent RNAs, histone mRNAs, long noncoding RNAs (lncRNAs), circular RNAs (circRNAs), and enhancer RNAs (eRNAs)[1–7]. Non-poly(A) RNAs are dynamically regulated and involved in important biological processes[3,4,8,9]. For example, measurement of eRNAs, which are mostly non-poly(A)[7,10], using total RNA-seq has provide insights into transcriptional regulation[6,7,8,10]. Second, total RNA-seq also has contributed to the detection of important RNA-processing events, including alternative and aberrant RNA splicing, and intron retention in cell differentiation and diseases[11]. Recently, multistep splicing (recursive splicing (RS), nested splicing, and re-splicing) has been discovered in vertebrates[12–14]. Based on the achievements made by total RNA-seq, development of a single-cell total RNA-seq method would be expected to help us fully investigate these molecular events in single cells.

Ideal single-cell total RNA-seq would have high sensitivity, especially to non-poly(A) RNAs, to fully capture transcriptome in single cells because the expression of lncRNAs and eRNAs, which are mainly non-poly(A)[15,16], is generally low[17]. Likewise, since pre-mRNAs generally contain intronic regions that are longer than exonic regions[1], unbiased amplification of full-length coverage along transcripts is essential to detect RNA-processing events such as co-transcriptional splicing and RS.

Thus far, much effort has been made to develop single-cell RNA-seq (scRNA-seq) methods with full-length coverage or sensitivity to non-poly(A) RNAs. Several scRNA-seq methods combining oligo-dT priming and template switching have been reported to provide full-length coverage of transcripts[18,19]. However, these methods are targeted at only poly(A) RNAs due to oligo-dT priming. Recently, SUPeR-seq, which employs specialized random primers conjugated to poly(T), was reported to detect non-poly(A) RNAs, including circRNAs[20]. Nonetheless, SUPeR-seq shows relatively low sensitivity with non-poly(A) RNAs (20–30%)[20], which leaves room for developing scRNA-seq methods with higher sensitivity to non-poly(A) RNAs. In addition, how to reduce the sequence derived from ribosomal RNAs (rRNAs) that accounts for most of the total RNA is a major task for establishing single-cell total RNA-seq. This issue is encountered because scRNA-seq uses a trace amount of total RNA as a template, and it is difficult to apply general rRNA-depletion methods that cause loss of RNA. Altogether, a single-cell total RNA-seq method with both full-length transcript coverage and high sensitivity for non-poly(A) RNAs remains to be developed.

scRNA-seq methods consist of various steps (Supplementary Fig. 1). The sensitivity of the method is the product of the efficiency of each step. Therefore, the number of steps and the sensitivity are inversely proportional. According to the conventional method, molecules that have not been captured in the reverse transcription (RT) step or have not been converted to the amplifiable form after the second-strand synthesis cannot become sequencing library DNA no matter how much the amplification method is improved. Therefore, it is necessary to simplify an experimental procedure, to use RT with high capture efficiency and to amplify the cDNA as early step as possible. To satisfy these conditions, we decided to use a novel RT technology: RT with random displacement amplification (RT-RamDA).

Herein, we developed the first, to our knowledge, full-length single-cell total RNA-seq method, which we named random displacement amplification sequencing (RamDA-seq), by combining RT-RamDA and not-so-random primers (NSRs). RT-RamDA provides global cDNA amplification directly from RNA during RT, which benefits RT efficiency, simplifies the procedure,

and decontaminates genomic DNA. NSRs enables random priming while preventing cDNA synthesis from rRNAs[21,22]. Using diluted total RNA, we confirm that RamDA-seq is single-cell total RNA-seq, which has high sensitivity to non-poly(A) and full-length coverage for extremely long transcripts exceeding 10 kb. We applied RamDA-seq to mouse embryonic stem cells (mESCs) undergoing differentiation and find cell state-dependent expression of known and novel non-poly(A) RNAs, including the extremely long non-poly(A) isoform of *Neat1*, a mammalian-specific lncRNA. Furthermore, we discover RS within >300-kb introns in single cells. Finally, RamDA-seq enables genome-wide analysis of eRNAs in single cells, which reveals the cell type-specific activity and potential regulators of the detected eRNAs. Our results suggest that RamDA-seq will provide insight into the dynamics of gene expression, transcriptional regulation, and RNA processing at the single-cell level.

## Results

**The principle and approach of RamDA-seq.** RamDA-seq consists of two fundamental techniques: a novel RT technology, RT-RamDA; and NSRs. First, RT-RamDA is a whole-transcriptome amplification (WTA) method that amplifies cDNAs directly from an RNA template (Hayashi et al., submitted). This method uses the nuclease activity of DNase I to introduce nicks in the cDNA and randomly displaced strands to amplify the cDNA during RT by RNase H minus reverse transcriptase. The use of the T4 gene 32 protein, a single-stranded DNA-binding protein, promoted strand displacement and protected the amplified cDNA against degradation by DNase I (Fig. 1a, Supplementary Fig. 1, and Supplementary Note 1 and 2). These events occurred continuously and consequently increased cDNA yields globally more than 10-fold (Fig. 1b) (Hayashi et al., submitted). Second, we used NSRs rather than N6 random primers to reduce the rRNA sequence[21,22]. NSRs are designed to avoid synthesizing cDNA from the rRNAs by removing 6-mers that exactly match the rRNA sequences from N6 random primers. Therefore, the use of NSRs enables the application of RT-RamDA for scRNA-seq.

We established the proper cell lysis conditions needed to expose nuclear-enriched non-poly(A) RNAs and remove genomic DNA before performing random priming-based RT (Methods section, Supplementary Fig. 2, and Supplementary Note 3). We also confirmed that the RamDA-seq protocol did not produce a library of DNA derived from genomic DNA or environmental DNAs (Supplementary Fig. 3).

We successfully established RamDA-seq on cell sorter-based microplate and Fluidigm C1 platforms, the latter of which is an automatic sample preparation system for single cells (RamDA-seq and C1-RamDA-seq, respectively; Fig. 1c).

**RamDA-seq shows high sensitivity and reproducibility.** To critically assess the performance of our method, we prepared a sequencing library from 10 pg of diluted mESC total RNA using RamDA-seq and C1-RamDA-seq. We then compared the performance of these methods with that of SUPeR-seq[20] and the following oligo-dT primer-based methods: SMART-Seq v4, a commercially available kit based on Smart-seq2[18,19]; and Quartz-Seq[23] (Supplementary Fig. 1). To provide upper limits for the analyses, we also prepared bulk rRNA-depleted total RNA-seq (rdRNA-seq) and poly(A)-selected RNA-seq (paRNA-seq) libraries using 1 μg of total RNA.

We calculated the number of detected transcripts that exhibited expression changes of less than twofold against rdRNA-seq. RamDA-seq and C1-RamDA-seq detected the largest number (~17 000 and 14 000, respectively) of transcripts among the scRNA-seq methods (Fig. 1d and Supplementary Fig. 4d). We

also plotted the squared coefficient of variation ($CV^2$) against expression levels to examine reproducibility. Compared with the other single-cell methods, RamDA-seq exhibited a lower degree of variation at all expression levels and more closely resembled the bulk RNA-seq methods (rdRNA-seq and paRNA-seq; Fig. 1e and Supplementary Fig. 4f). Furthermore, RamDA-seq and C1-RamDA-seq showed higher correlation in expression level

with rdRNA-seq than did the other scRNA-seq methods (Supplementary Fig. 4h), indicating that RamDA-seq is highly similar to rdRNA-seq. Read distributions, especially high proportions of intronic, 5′ untranslated region and intergenic regions, showed that RamDA-seq resembled rdRNA-seq, whereas SUPeR-seq was more similar to paRNA-seq and the oligo-dT primer-based methods (Supplementary Fig. 4i). We also

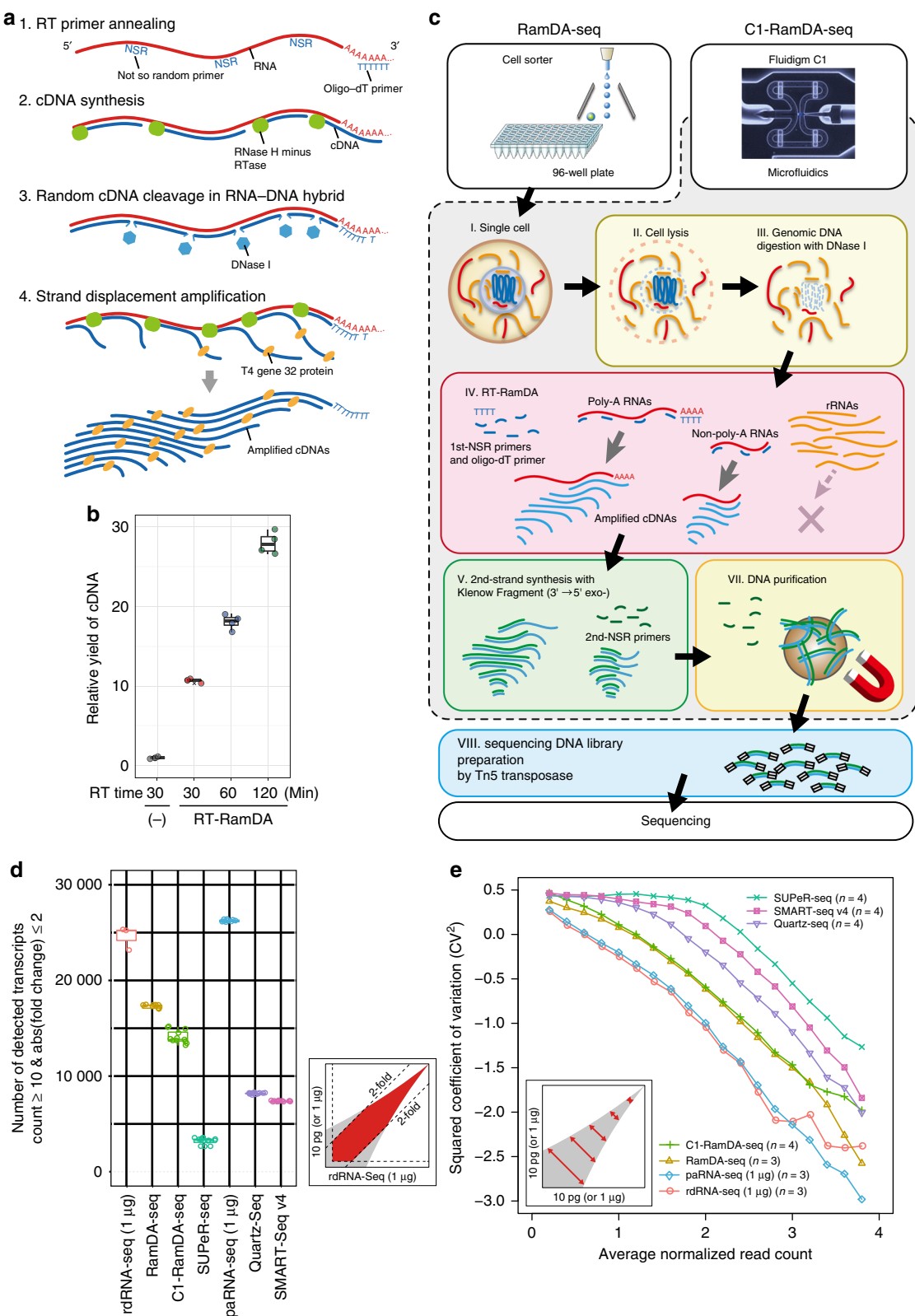

compared large-volume-inputted RamDA-seq with rdRNA-seq using 1 ng of RNA. When we compared RamDA-seq and rdRNA-seq with the same input amount, there was hardly any difference in sensitivity and reproducibility except for the contamination rate of rRNA (Supplementary Fig. 5). Finally, we carried out a spike-in RNA experiment; the results clearly indicated that RamDA-seq was highly reproducible and sensitive (Supplementary Note 4 and Supplementary Fig. 6). Altogether, based on the substantial resemblance of RamDA-seq to bulk total RNAs-seq, we conclude that RamDA-seq is a robust, single-cell total RNA-seq method.

**RamDA-seq shows full-length coverage of extremely long RNA.** To evaluate the transcript coverage of the sequence reads in detail, we compared the percentage coverage with the absolute length of the transcripts by using 10 pg of RNA (Fig. 2a and Supplementary Fig. 7). Remarkably, RamDA-seq, similar to bulk RNA-seq methods, showed full-length coverage, whereas other scRNA-seq methods did not. Consistently, we also confirmed that the read coverage of RamDA-seq against relative transcript length was most similar to that of rdRNA-seq (Supplementary Fig. 8d). Specifically, the coverage of SMART-Seq v4 markedly decreased in the 10- to 20-kb range (Fig. 2a), even though this method employs template switching, which has been reported to provide full-length transcript coverage[18,19]. This result was consistent with previously reported results[24,25]. For example, mapping data to *Mdn1* (17 970 bp) revealed missing exons in the middle range of the transcript when using SMART-Seq v4, whereas complete mapping to *Mdn1* was achieved when using RamDA-seq, similar to rdRNA-seq (Fig. 2b). Similar differences in mapping data were also observed for other long (>10 kb) transcripts in both 10 pg of RNA and single cells (Supplementary Fig. 9 and 10). In addition, the fraction of exonic regions covered by the reads indicated that RamDA-seq covered a higher fraction of exonic regions than did the other methods in all length bins (Fig. 2c and Supplementary Fig. 8a–c). These results indicate that RamDA-seq can provide full-length coverage even for extremely long (>10 kb) transcripts.

**RamDA-seq shows high sensitivity with non-poly(A) RNA.** We next asked whether RamDA-seq could detect non-poly(A) RNAs. First, we evaluated whether RamDA-seq could detect the expression of histone-coding genes, well-known non-poly(A) RNAs, using 10 pg of RNA data from mESCs. RamDA-seq detected more histone-coding genes than did the other scRNA-seq methods, including SUPeR-seq, which is reported to detect non-poly(A) RNA[20] (Fig. 2d). We further confirmed that RamDA-seq could quantitatively detect oscillation in expression levels of histone mRNAs through the cell cycle in mESCs at the single-cell level (Supplementary Fig. 11; see Supplementary Note 5 for further discussion).

To systematically evaluate the detection performance of RamDA-seq for non-poly(A) RNAs, we first identified non-poly(A) RNA candidates expressed in mESCs using bulk total and poly(A) RNA-seq data (811 and 7935 for strict and loose criteria, respectively; Methods section). RT-quantitative PCR (RT-qPCR) analyses confirmed that these candidates were indeed non-poly(A) RNAs (Supplementary Fig. 12). We then compared the performance of scRNA-seq methods for detecting these sets of non-poly(A) RNAs. RamDA-seq detected the highest number of non-poly(A) transcripts among the scRNA-seq methods (Supplementary Fig. 13a), which was true even for lowly expressed non-poly(A) transcripts (Fig. 2e). In addition, the correlation of the expression levels with bulk total RNA-seq was higher for RamDA-seq than for the other scRNA-seq methods (Supplementary Fig. 13b,c). These results confirm that RamDA-seq has high sensitivity with non-poly(A) RNAs.

**Cell state-dependent non-poly(A) RNA in single cells.** To test whether RamDA-seq could be used to measure the expression profiles of non-poly(A) RNAs in biological samples, we performed RamDA-seq with mESCs collected 0, 12, 24, 48, and 72 h after the induction of cell differentiation into primitive endoderm (PrE) cells (Supplementary Fig. 14b, c). We first confirmed that RamDA-seq could specifically detect the expression of differentially expressed non-poly(A) transcripts, which were identified by bulk RNA-seq, at the single-cell level (Supplementary Note 6 and Supplementary Fig. 13d).

Diffusion map analysis revealed variability within cells even at the same time points (Fig. 3a), and we searched for transcripts whose expression levels changed with pseudotime (using the first diffusion component (DC1) as pseudotime; see Methods section). We identified 7580 such transcripts, including 458 non-poly(A) transcripts (Fig. 3b), and divided the 7580 transcripts into seven clusters based on expression patterns (Fig. 3c). The clustering results were supported by the expression patterns of ES and PrE marker genes (Fig. 3c and Supplementary Note 7). The dynamically regulated non-poly(A) transcripts were spread in all clusters with various expression patterns, suggesting that non-poly(A) transcripts are involved in various cell functions. Using single-cell preamplification RT-qPCR (scRT-qPCR), we validated the observed expression changes in several of the non-poly(A) transcripts, including two unannotated intergenic non-poly(A) transcripts (clusters 1 and 2) and *Hist1h1a* (cluster 5; Fig. 3d and Supplementary Fig. 15g). Furthermore, reasoning that transcripts

**Fig. 1** Overview of RT-RamDA and single-cell RamDA-seq. **a** Schematic diagram of RT-RamDA. 1. RT primers (oligo-dT and not-so-random primers) anneal to a RNA template. 2. Complementary DNA (cDNA) is synthesized by the RNA-dependent DNA polymerase activity of RNase H minus reverse transcriptase (RTase). 3. Endonuclease (DNase I) selectively nicks the cDNA of the RNA:cDNA hybrid strand. 4. The 3′ cDNA strand is displaced by the strand displacement activity of RTase mediated by the T4 gene 32 protein (gp32), starting from the nick randomly introduced by DNase I. cDNA is amplified as a displaced strand and protected by gp32 from DNase I. **b** Relative yield of cDNA molecules using RT-qPCR (*n* = 4). Mouse ESC total RNA (10 pg) was used as a template, and 1/10 the amount of cDNA was used for qPCR. The relative yield was calculated by averaging the amplification efficiency of four mESC (*Nanog*, *Pou5f1*, *Zfp42*, and *Sox2*) and three housekeeping (*Gnb2l1*, *Atp5a1*, and *Tubb5*) genes using a conventional method (−) as a standard. **c** Schematic diagram of RamDA-seq and C1-RamDA-seq. For details, please refer to the Methods section. **d** Number of detected transcripts with twofold or lower expression changes against rdRNA-seq (count ≥ 10). For the boxplots in **b** and **d**, the center line, and lower and upper bounds of each box represent the median, and first and third quartiles, respectively. The lower (upper) whisker extends to smallest (largest) values no further than 1.5 × interquartile range (IQR) from the first (third) quartile. **e** Squared coefficient of variation of the read count. All conditions were adjusted, and 10 million reads were used in **d** and **e**. Transcripts were annotated by GENCODE gene annotation (vM9)

with similar expression patterns should share biological functions, we attempted to infer the potential functions of these dynamically regulated non-poly(A) transcripts by performing functional enrichment analysis of each cluster (Supplementary Data 1; see Supplementary Note 8 for further discussion). Future studies of these non-poly(A) RNAs will enhance our understanding of ESC differentiation.

We next focused on the long isoform *Neat1-001* (*Neat1_2*), which is required for the formation of paraspeckles[26]. The expression level of *Neat1-001* specifically decreased at 12 h in

RamDA-seq and scRT-qPCR (Fig. 3e). Since *Neat1-001* is a super-long (>20 kb) non-poly(A) lncRNA, we thought it would be an optimal indicator of sensitivity for non-poly(A) RNAs and full-length coverage. Consistently, mapping data of single cells using RamDA-seq showed full-length transcript coverage for *Neat1-001* (Fig. 3e). *Neat1* also has a polyadenylated short isoform (*Neat1-002*; *Neat1_1*) that is transcribed from the same promoter as *Neat1-001*[27]. To assess whether the observed decrease was specific to the long isoform or common to the two isoforms, we compared the read coverage of the region

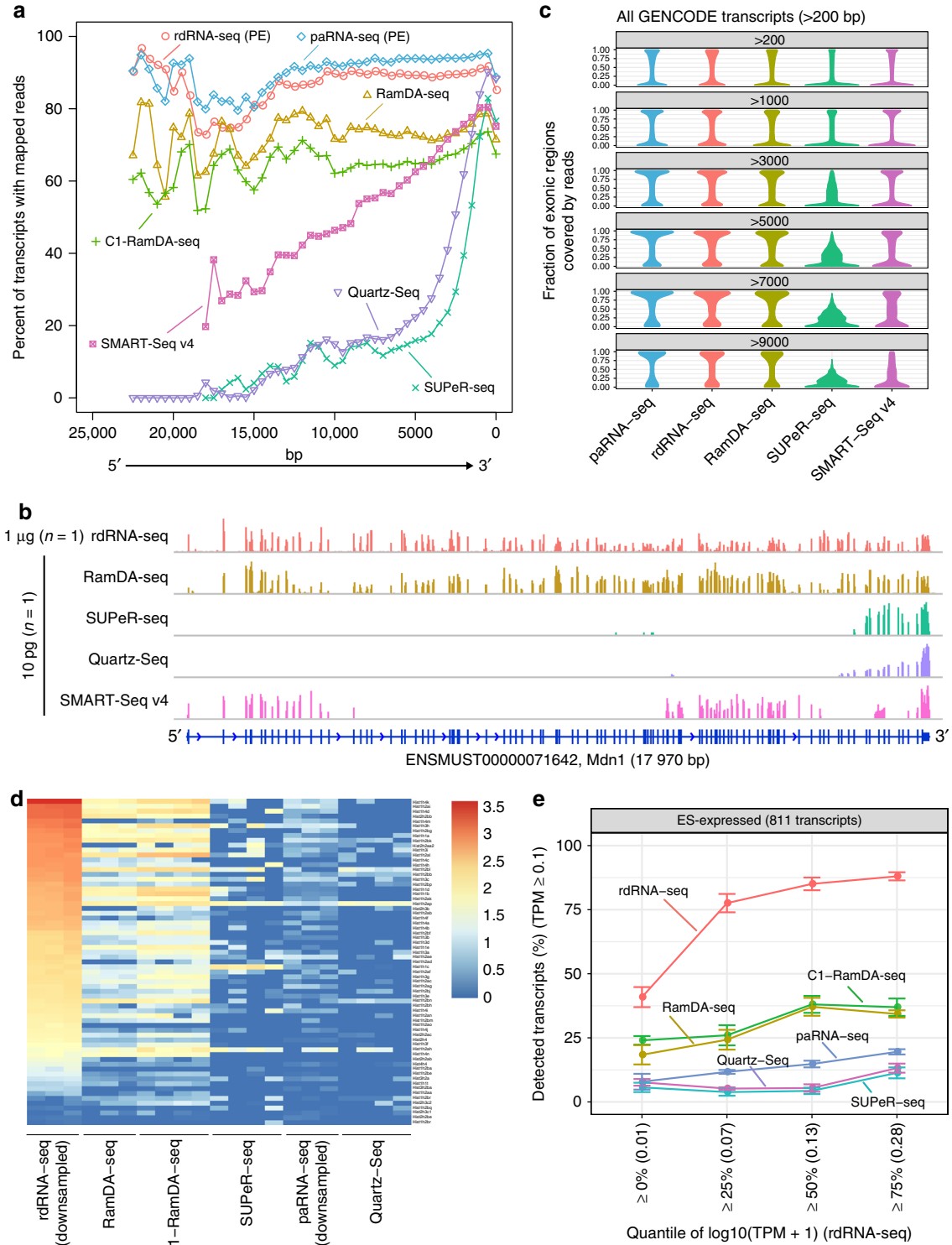

common to both isoforms and the region specific to the long isoform. The expression of the long isoform was significantly lower than that of the short isoform at 12 h only (Fig. 3f; Wilcoxon signed-rank test, *p*-value < 0.001 after the Bonferroni correction). Further studies are necessary to elucidate the potential biological significance of the observed dynamics of *Neat1* isoforms.

Collectively, these results indicate that many non-poly(A) transcripts are dynamically regulated and highlight the utility of full-length single-cell total RNA-seq for studying the dynamic regulation and potential functions of non-poly(A) RNAs.

**Recursive splicing in single cells.** Total RNA-seq can detect pre-mRNAs and thereby confers the ability to investigate the dynamics of RNA-processing events[1], including RS[13]. RS is a multistep process of intron removal using cryptic splice sites within long introns[14] and was recently observed in vertebrates[13]. Due to the large number of intronic reads (Supplementary Fig. 4i) and high sensitivity of RamDA-seq, we hypothesized that RamDA-seq could detect RS. If RS occurred, read coverage patterns similar to a sawtooth wave would be observed, with a gradual decrease from the 5′ to the 3′ end of the long intron[13] (Fig. 4a). We searched for transcripts with a sawtooth pattern within long introns by fitting linear regression models against RamDA-seq read coverage averaged across all cells in the differentiation time-series data ($p < 1e-5$, Methods section). We found clear patterns of RS in *Cadm1*, *Robo2*, and *Magi1*, which were all previously reported to show RS[13] but were first reported in mESCs. For example, *Cadm1* and *Robo2* showed clear sawtooth patterns as well as novel splicing junctions in RamDA-seq and bulk total RNA-seq data (Fig. 4b, c). The averaged RamDA-seq read coverage for each time point showed that the sawtooth wave patterns persisted at all time points for *Cadm1* and *Magi1*, whereas the pattern was observed at only 48 and 72 h for *Robo2* (Fig. 4d, e and Supplementary Fig. 16). The height of the sawtooth wave pattern was associated with the expression level of host genes at all time points (Supplementary Fig. 16), while the pattern was not observed when gene expression was hardly detected (Supplementary Fig. 16). A simulation-based estimation of the sensitivity of RS detection showed that RS was robustly detected when host genes were sufficiently expressed (transcript per million (TPM) >1), corroborating the above observations (Methods section; Supplementary Fig. 16, right panels).

Next, we attempted to address whether RamDA-seq can detect RS even in each single cell. Based on the RS detection sensitivity estimated above, we selected cells with sufficient intronic reads (RS detection probability >0.95; Supplementary Fig. 17, Methods section). We fitted linear regression models against the RamDA-seq read coverages of each single cell in *Cadm1*, *Robo2*, and *Magi1*. RS was detected in a subpopulation of cells (71 of 149 cells

for *Cadm1*, 12 of 54 cells for *Magi1*, and 1 of 1 cell for *Robo2*) although many cells in which RS was not detected also appeared to show the sawtooth pattern (Supplementary Fig. 17). However, interestingly, some other cells showed monotonically decreasing patterns, which correspond to "normal" splicing (Fig. 4a). The monotonically decreasing pattern was also observed even when we filtered cells with a more stringent threshold of intronic reads (Supplementary Fig. 17). These observations raise the possibility of cell-to-cell heterogeneity in RS. Therefore, further investigation is needed to reveal the mechanisms and significance of the observed heterogeneity in RS. Collectively, these results demonstrate that RamDA-seq can detect RS in single cells.

**Enhancer RNAs in single cells.** Most eRNAs represent one major class of non-poly(A) transcript[7,10], and previous studies have often used bulk total RNA-seq methods to detect the expression of eRNAs[28,29]. Although compared with mRNAs, eRNAs are expressed at quite low levels[16], we hypothesized that the high sensitivity of RamDA-seq to non-poly(A) RNAs could allow the detection of eRNAs in single cells. To address this possibility, we first assessed the performance of RamDA-seq and other scRNA-seq methods for detecting eRNAs using two sets of eRNAs: (1) the transcribed enhancer annotation in mESCs identified using CAGE (cap analysis of gene expression) by the FANTOM5 project[10]; and (2) the non-poly(A) RNAs with their transcription start site (TSS) displaying enhancer-like histone modifications (Methods section). Analysis of the 10 pg of RNA data confirmed that RamDA-seq could detect eRNAs with higher sensitivity than could the other scRNA-seq methods (Supplementary Note 9 and Supplementary Fig. 18).

Next, we examined whether RamDA-seq could be used to detect eRNAs in single cells with differentiation time-series data. The detection rates for ES-active CAGE enhancers were highest at 0 h and decreased as differentiation progressed (Fig. 5a), whereas the detection rates for ES-inactive enhancers were consistently low. Similar trends were observed when we used non-poly(A) eRNAs with enhancer-like histone modifications (Supplementary Fig. 19a). In addition, we checked the distribution of read coverage around ES-active CAGE enhancers. Bimodal peaks were observed in regions 200–400 bp away from the enhancers in cells at 0 h (Fig. 5b and Supplementary Fig. 19b,c), as observed in the 10 pg of RNA data (Supplementary Fig. 18b,c), and these peaks weakened with the progression of cell differentiation. Consistently, bimodal peaks are observed around enhancers in the read coverage of total RNA-seq[7]. On the other hand, the distribution of the read coverage around random genomic regions was steadily low across all time points (Fig. 5b). Collectively, these results indicated that RamDA-seq could detect eRNAs in a cell type-specific manner.

**Fig. 2** Read coverage across transcripts and non-poly(A) RNA detection using scRNA-seq methods. **a** Percentage of sequence read coverage throughout the transcript length. The x-axis shows the absolute distance (bp) from the 3′ end of the transcripts ($x_i$). The y-axis shows the fraction of transcripts with read coverage ($n_i/N_i$). $n_i$: the number of transcripts of which at least one read was mapped within the range of ($x_i$, $x_{i+1}$). $N_i$: the number of transcripts with $\geq x_i$ transcript length. Only transcripts in the GENCODE (vM9) annotations with transcript per million (TPM) $\geq 1$ in rdRNA-seq results and with $\geq 200$-bp transcript length were considered. PE: data from paired-end reads. **b** Visualization and comparison of mapped reads of a long transcript, *Mdn1* (17 970 bp). We selected *Mdn1* as the gene with the highest number of exons (102 exons) in the 25 genes with length $\geq 10$ kb and TPM $\geq 5$ in rdRNA-seq results. **c** Distribution of the fraction of exonic regions covered by sequenced reads with 10 pg of RNA data for all transcripts with >200-bp transcript length in the GENCODE (vM9) annotations. The transcripts were sorted into bins (represented by the number at the top of each panel) according to transcript length. **d** The sensitivity for detecting histone transcripts using 10-pg RNA samples. Each row represents a histone transcript. Each column represents a sample using the indicated scRNA-seq method. The expression levels in log10 (TPM + 1) quantified by sailfish are indicated according to the color key. **e** Detection rates of non-poly(A) transcripts (strict criterion) expressed in ESCs for different expression level thresholds in rdRNA-seq. The points and error bars represent means and SDs, respectively. Each line represents a scRNA-seq method. The numbers in parentheses represent the number of transcripts

Previous studies have demonstrated an enrichment for cell type- and condition-specific transcription factor DNA-binding motifs at active eRNA loci[10,30], which prompted us to search for enrichment of motifs of cell type-specific transcription factors. We defined CAGE enhancer subsets that were active in ESCs using RamDA-seq and performed a motif enrichment analysis

(Methods section). In parallel, the same analysis was performed using rdRNA-seq. RamDA-seq identified 1515 ES-active enhancers, 75% of which were also considered active using bulk total RNA-seq (Fig. 5c). One hundred motifs were enriched ($q$-value < 0.05). Of these, 97 (97%) were also enriched based on the bulk total RNA-seq analysis. Interestingly, these enriched

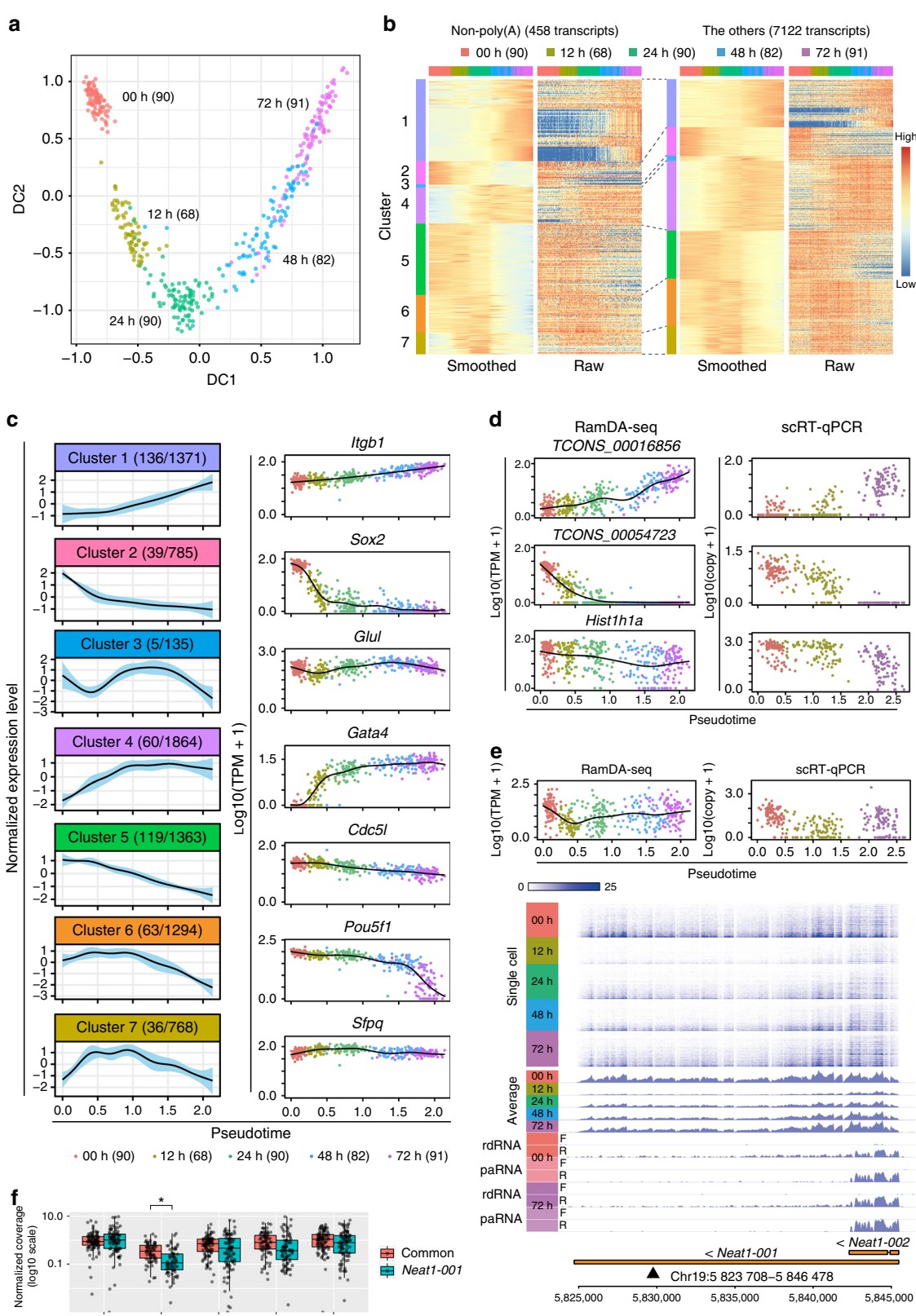

motifs contained motifs of transcription factors involved in self-renewal and ESC pluripotency[31,32], including motifs for KLF4, KLF5, SOX2, and n-MYC and a composite motif for "OCT4-SOX2-TCF-NANOG" (Fig. 5c).

Having validated our genome-wide eRNA analysis with RamDA-seq, we searched for eRNAs showing variations according to pseudotime of the cells and performed hierarchical clustering (Methods section). We found 1338 dynamically regulated eRNAs (false discovery rate (FDR) < 0.01) in five clusters: two downregulated clusters; two transiently upregulated clusters; and one late upregulated cluster (Fig. 5d, e). Fewer eRNAs were found in the late upregulated clusters, possibly due to hyperactive global transcription in ESCs[33] or the lack of PrE data in the FANTOM annotation. Notably, GATA4, a late PrE marker[34], was enriched in one transiently upregulated cluster (cluster 3) and in the late upregulated cluster (cluster 5), which suggests that these clusters represent enhancers that function in the differentiation into PrE. Altogether, we conclude that RamDA-seq can detect eRNA activity associated with cell type-specific regulation as well as the potential regulator of eRNAs within a subpopulation of single cells.

## Discussion

In this study, we developed RamDA-seq, a full-length total RNA sequencing of single cells, and showed that RamDA-seq was the most effective at detecting full-length and non-poly(A) transcripts among the scRNA-seq methods tested here (Fig. 2). RamDA-seq revealed many known and unannotated non-poly(A) transcripts that were dynamically regulated as differentiation progressed, including Neat1-001 (Fig. 3). In addition, RamDA-seq detected RS (Fig. 4) and eRNAs (Fig. 5); this is, to our knowledge, the first report of genome-wide analysis in single cells. Our results demonstrate that RamDA-seq provides a comprehensive view of total RNA, including non-poly(A) RNAs, pre-mRNA, and eRNA, at the single-cell level.

The sensitivity and full-length transcript coverage of RamDA-seq were achieved using RT-RamDA and NSRs. RT-RamDA contributes to robustness to template loss via manipulation by cDNA amplification during RT, which leads to high sensitivity. Moreover, RT-RamDA improves sensitivity and reproducibility by eliminating the necessity for PCR amplification, which often results in amplification bias. NSRs contributes to the full-length transcript coverage and high efficiency of capturing poly(A) and non-poly(A) RNAs by multiple priming. RT-RamDA is suited for the use of NSRs because RT-RamDA can amplify cDNA without the adapter sequences for WTA, unlike other conventional methods (Supplementary Fig. 1). These characteristics contribute to RT efficiency and the cost reduction of oligo primer synthesis.

There are some limitations to this method. Since RT-RamDA generates cDNA by strand displacement amplification, which depends on the random nicking of cDNA, cell barcodes and unique molecular identifiers (UMIs) could not be added to the sequencing library. Therefore, it is difficult to perform pre-indexing high-throughput sequencing and molecule counting using UMIs. Moreover, because RamDA-seq is a total RNA-seq method, it requires more sequencing reads than do other scRNA-seq methods (described below). Even though RamDA-seq uses NSRs, the RamDA-seq library still contains a relatively high proportion (10–35%) of rRNA sequences (Supplementary Fig. 4b, c and Supplementary Fig. 15c). To address this issue, modifying the NSRs is necessary, for example, by adjusting the annealing temperature of NSRs to prevent misannealing or removing the complementary sequences annotated as rRNAs in RepeatMasker. It is also important to achieve strand-specific sequencing in RamDA-seq to distinguish overlapping transcripts.

Based on a subsampling simulation, with >1M reads per cell, RamDA-seq detects more transcripts than the other scRNA-seq methods (Supplementary Fig. 4e). Moreover, with ~4M reads per cell, RamDA-seq yields reads from non-poly(A) transcripts, introns and intergenic regions and provides beneficial information regarding unannotated intergenic transcripts, RS, and enhancer RNA (Figs. 3–5). Given that ~4M reads per cell are typical for plate-based scRNA-seq (for example, 96 cells in 1 run on NextSeq yields ~4M reads per cell), these results demonstrate that RamDA-seq needs just normal sequencing runs to provide useful information regarding gene expression, transcriptional regulation, and RNA processing.

Full-length total RNA-seq from single cells will be valuable to many studies using rare cells. Many biologically and clinically important cell types are rare and are often found in heterogeneous cell populations. Thus, these cell types require single-cell approaches, and accumulating evidence suggests the importance of full-length total RNA-seq in single cells. Non-poly(A) lncRNAs[35] and circRNAs[36], non-canonical splicing[11], fusion genes[36], mutations[36], and RNA editing[37] are associated with many diseases, such as cancers, and their detection will benefit from full-length coverage of mRNA and pre-mRNAs. Enhancers account for cell type-specific expression[28,29] and diseases, and their activity and potential regulators can be inferred by eRNAs[10,30]. Based on our results, single-cell analyses using RamDA-seq could be useful for identifying novel biomarkers and drug targets, non-canonical and aberrant RNA-processing events, and active enhancers and their potential regulators in rare cells.

---

Fig. 3 RamDA-seq analyses of cell differentiation. **a** A diffusion map of the cells collected over time and colored by the sampling time points. DC, diffusion component. The numbers in parentheses represent the number of cells. **b** Heat maps of the expression levels of non-poly(A) (left) and the other (right) transcripts. Rows are ordered and colored by clusters. Columns are ordered by pseudotime and colored by sampling time points. Smoothed values are transformed to Z-scores for each row. Raw values are scaled from 0 to 1 for each row. **c** (Left) Averaged expression profile for each cluster. The x-axis represents pseudotime. Thin, colored areas represent SDs. The numbers before and after the slash in the parenthesis represent the numbers of non-poly (A) transcripts and all transcripts included in each cluster, respectively. (Right) Expression profile of the representative transcript for each cluster. Each black curve represents a fitted generalized additive model. **d** Expression profiles of two unannotated intergenic non-poly(A) transcripts measured by RamDA-seq (left) and single-cell preamplification RT-qPCR (scRT-qPCR) (right). **e** (Top) Expression profiles of Neat1-001 measured by RamDA-seq (left) and scRT-qPCR (right). The x-axes represent pseudotime. (Bottom) Coverage plot of RamDA-seq at the Neat1 locus. The upper heat map represents the read coverage at the single-cell level. The middle plots represent the coverage averaged for cells at each time point as well as those of rdRNA-seq and paRNA-seq. Gene models are shown at the bottom. The arrowhead indicates the position of the qPCR primer. **f** The read coverage of the region common to both Neat1-001 and Neat1-002 (common) and the region specific to Neat1-001. The read coverage was normalized to the average of all cells. The asterisk indicates significant difference between two regions (Wilcoxon signed-rank test, p-value < 0.001 after Bonferroni correction). The center line, and lower and upper bounds of each box represent the median, and first and third quartiles, respectively. The lower (upper) whisker extends to smallest (largest) values no further than 1.5 × IQR from the first (third) quartile

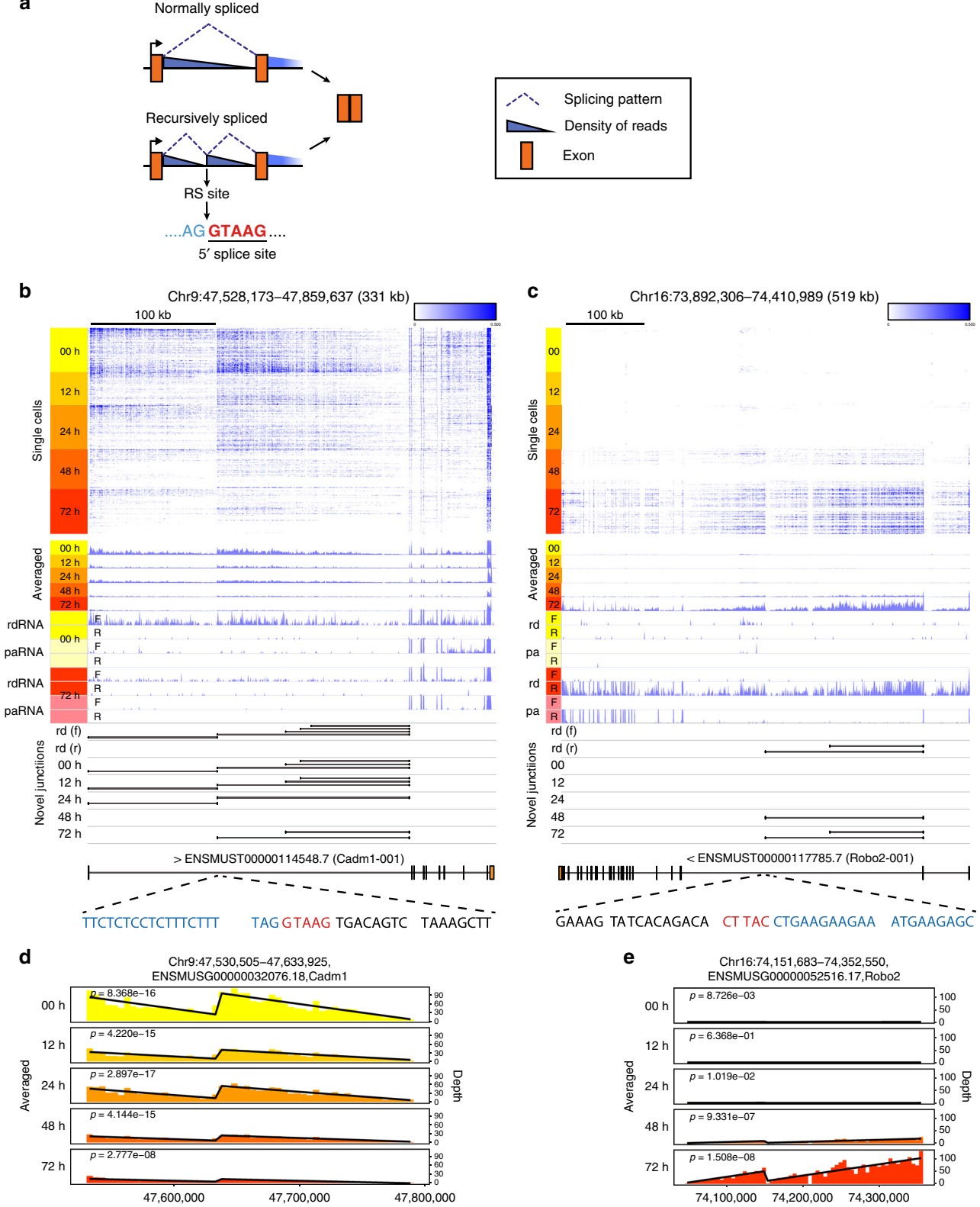

**Fig. 4** Single-cell analysis of recursive splicing. **a** Illustration of the expected read coverage for recursive-splicing sites. **b**, **c** Recursive-splicing exons observed by RamDA-seq in *Cadm1* (**b**) and *Robo2* (**c**). The upper heat maps represent the RamDA-seq read coverage for each cell. The middle tracks represent the averaged RamDA-seq coverage for each sampling time point. The lower tracks represent the read coverage of rdRNA-seq (rd) and paRNA-seq (pa) on forward (f) and reverse (r) strands at 0 and 72 h. Novel splice junctions observed by rdRNA-seq (rd) on forward (f) and reverse (r) strands and RamDA-seq are also shown. Gene models and nucleotide sequences around recursive-splicing sites are shown at the bottom. The region upstream (blue) of the 5′ splice motif (red) has been excised to reconstitute the 5′ splice site. **d**, **e** The summed normalized read coverage of RamDA-seq for each time point in the 5-kb bin (bars) and the fitted linear regression models (black lines) in *Cadm1* (**d**) and *Robo2* (**e**). The *p*-values of F-tests are indicated

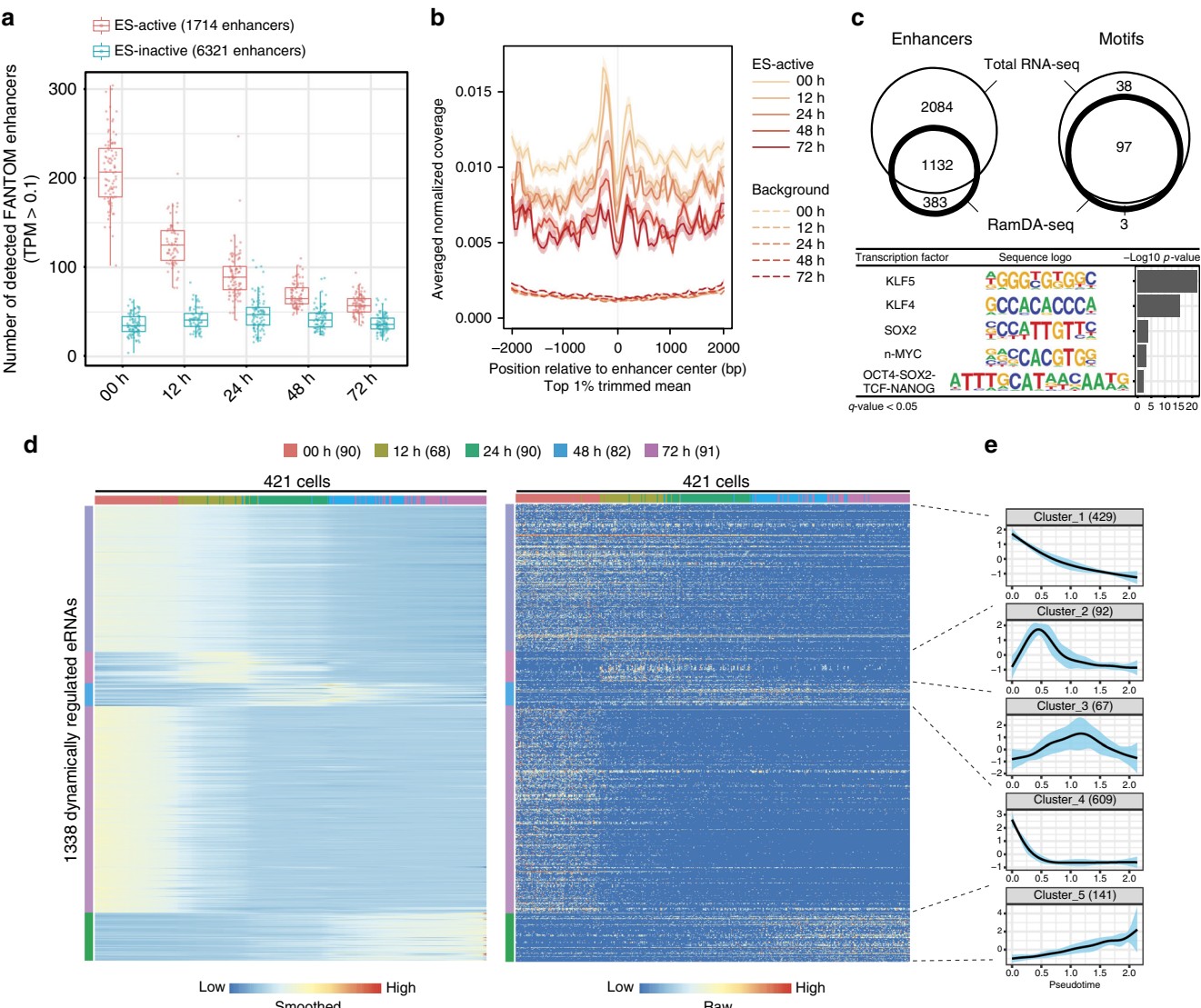

**Fig. 5** Single-cell analysis of enhancer RNAs. **a** Number of detected FANTOM5 CAGE enhancers for the ES-active (red) and ES-inactive (blue) enhancers for each cell. The detection of each enhancer is called when the TPMexceed 0.1. The center line, and lower and upper bounds of each box represent the median, and first and third quartiles, respectively. The lower (upper) whisker extends to smallest (largest) values no further than 1.5 × IQR from the first (third) quartile. **b** Aggregation plot of the read coverage around the ES-active enhancers (solid lines; 1199 enhancers) and random genomic regions (dashed lines; 44 467 enhancers). Normalized read coverage was averaged across the cells and normalized to make the sum of each enhancer be 1 and were further averaged across enhancers after trimming the top 1% of enhancers for each 50-bp bin. Only the top 70% of highly expressed enhancers (quantified within regions 75 to 275 bp away from the center of enhancers) were included. The shaded areas represent SDs. **c** Motif analysis of enhancers detected by rdRNA-seq and RamDA-seq with ESCs. The number of detected enhancers (top left) and enriched known DNA motifs of transcription factors (top right) are shown in Venn diagrams. Sequence logos and p-values for enrichment (binomial test) are indicated at the bottom for overrepresented motifs (q-value < 0.05) of transcription factors associated with self-renewal in ESCs. **d** Heat maps of the expression levels of eRNAs. Rows are ordered and colored by clusters. Columns are ordered by pseudotime and colored by sampling time points. Smoothed values are transformed to Z-scores for each row. Raw values are scaled from 0 to 1 for each row. The numbers in parentheses represent the number of cells. **e** Averaged expression profile for each cluster of eRNAs. The x-axis represents pseudotime. Thin, colored areas represent SDs. The numbers in parentheses represent the number of eRNAs

*Neat1* is an architectural component of paraspeckle nuclear bodies[38], which regulate gene expression via capture of A-to-I edited mRNAs[39] and transcription factors[40], and is required for corpus luteum formation and establishment of pregnancy in mice[41]. The long non-poly(A) isoform *Neat1-001*, not the short poly(A) isoform *Neat1-002*, is essential for the formation of paraspeckles[42]. Although the two isoforms are transcribed from the same promoter, they show different expression patterns, and *Neat1-001* is expressed only in a small subpopulation of cells in adult mouse tissues[26]. Therefore, distinguishing the expression of

the two isoforms of *Neat1* at the single-cell level is critical for studying their functions. In this study, RamDA-seq's sensitivity to non-poly(A) transcripts and full-length transcript coverage distinguished the expression of the two isoforms and revealed dynamic and differential regulation of the long non-poly(A) isoform (Fig. 3). These results suggest that RamDA-seq could be beneficial for investigation of temporal and spatial expression patterns of long non-poly(A) RNAs in single cells.

Unexpectedly, we observed cell-to-cell heterogeneity in read coverage patterns around RS sites, suggesting that some cells

showed RS, and other cells showed normal splicing (Supplementary Fig. 17). These results indicate that RamDA-seq can detect cell-to-cell heterogeneity in RS and could help to address the mechanisms and relationship between transcription and splicing. Toward these goals, several important challenges remain. Given that some cells in which RS was not detected showed weak sawtooth patterns, RamDA-seq highlights the limitation of the current linear regression model used to detect RS in this study and the need for further improvement in computational methods to robustly detect RS using single-cell data. Another challenge is to experimentally and computationally distinguish biological and technical variabilities in RS at the single-cell level. We will address these challenges in the future.

Recently, droplet-based scRNA-seq methods, which can sequence a very high number of cells at once, have been proposed[43,44]. Although a large number of sequenced cells benefits the discovery of rare cell populations[45], these methods, in contrast to RamDA-seq, target only the 3′ ends of poly(A) transcripts. Due to this difference, RamDA-seq will complement these methods for studying single cells in complex biological systems.

We confirmed that RamDA-seq works with living single cells on the Fluidigm C1 platform (Supplementary Fig. 14d and Supplementary Fig. 15a–e), which suggests that RamDA-seq can be easily applied in diverse studies. We also evaluated plate-to-plate variability (batch effect) of RamDA-seq (Supplementary Fig. 20a). The variabilities between plates were similar to the variabilities within plates, suggesting that the plate effects are modest at worst. Plate-to-plate variability was much smaller than the variability between cell types, and the proportion of variance explained in principal component analysis (PCA) was 0.6% (Supplementary Fig. 20b–h). In conclusion, we propose that RamDA-seq will expand our scope of single-cell measurement and will be useful for investigating cellular and transcriptional dynamics at the single-cell level in heterogeneous cell populations, such as cancers and complex tissues.

## Methods

**Cell culture**. 5G6GR mouse ES cells were used for total RNA extraction and scRNA-seq. 5G6GR mouse ES cells were provided by Hitoshi Niwa (Laboratory for Stem Cell Biology, Institute of Molecular Embryology and Genetics at Kumamoto University). This cell line was constructed by randomly incorporating the linearized Gata6-GR-IRES-Puro vector into EB5 ES cells[46]. The cells were cultured in a feeder-free gelatin-coated dish in 10% fetal calf serum containing Glasgow minimal essential medium (Sigma-Aldrich), 1000 U/mL leukemia inhibitory factor (ESGRO; Millipore), 100 μmol/L 2-mercaptoethanol (Thermo Fisher), 1× non-essential amino acids (Thermo Fisher), 1 mmol/L sodium pyruvate (Thermo Fisher), 2 mmol/L L-glutamine (Sigma-Aldrich), 0.5× penicillin/streptomycin (Thermo Fisher), 0.5 μg/mL puromycin (Sigma-Aldrich), and 10 μg/mL blasticidin (Thermo Fisher). To assess PrE differentiation, 5G6GR ES cells were cultured in differentiation medium containing 100 mmol/L dexamethasone rather than blasticidin. The cells were cultured for 72 h.

**Cell dissociation and single-cell sorting**. The cultured cells were dissociated with 1/5× TrypLE Express (Thermo Fisher) at 37 °C for 3 min. The dissociated cells were adjusted to 1 × 10⁶ cells/mL and stained in 10% Hoechst 33342 dye (Sigma-Aldrich) in phosphate-buffered saline (PBS) at 37 °C for 15 min to identify the cell cycle. After Hoechst 33342 staining, the cells were washed once with PBS and stained with 1 μg/mL propidium iodide (PI, Sigma-Aldrich) to remove dead cells. Single-cell sorting was performed as previously reported[23] using MoFlo Astrios (Beckman Coulter). Single cells were collected in 1 μL of cell lysis buffer in a 96-well PCR plate (BIOplastics). The data were analyzed using FlowJo 8.1 software (FlowJo).

**RNA extraction**. Total RNA was extracted using an RNeasy Mini Kit (Qiagen). RNA quantification and quality checks were performed with a Quantus fluorometer (Promega) and the Bioanalyzer RNA 6000 Nano Kit (Agilent Technologies). We confirmed that the sample RNA integrity number was >9.5.

**Bulk RNA-seq**. For paRNA-seq, we purified poly(A) RNA from 1 μg of ES total RNA using a NEBNext Poly(A) mRNA Magnetic Isolation Module (NEB). For rdRNA-seq, we depleted rRNA from 1 μg of ES total RNA using a GeneRead rRNA Depletion Kit (Qiagen). We prepared conventional RNA-seq library DNA with the resulting RNA using a commercial kit (NEBNext Ultra Directional RNA Library Prep Kit for Illumina; NEB) in accordance with the manufacturer's protocol, with slight modifications during RT and PCR steps. We used SuperScript III (Thermo Fisher) for RT. We used KAPA HiFi DNA polymerase (Kapa Biosystems) in the PCR step. When we performed rdRNA-seq using 10 or 1 ng of ES total RNA, we utilized a KAPA RNA HyperPrep Kit with RiboErase (Kapa Biosystems) for the preparation of sequencing library DNA according to the manual instructions.

**Cell lysis buffer conditions**. Total RNAs or sorted single cells were diluted or lysed in 1 μL of cell lysis buffer containing 1 U RNasein plus (Promega), 10% RealTime ready Cell Lysis Buffer (Roche), 0.3% NP40 (Thermo Fisher), and RNase-free water (TaKaRa). The lysate solution was immediately centrifuged and mixed using a ThermoMixer C (Eppendorf) at 2000 rpm for 1 min at 4 °C. The cell lysate solution was stored at −80 °C until use.

**RT-RamDA**. Template RNA was diluted in 1 μL of lysis buffer, denatured for 1.5 min at 70 °C, and stored on ice. The reaction buffer for RT was modified using the PrimeScript RT reagent Kit (TaKaRa). A mixture containing 2 μL of conventional RT mix (1.5× PrimeScript buffer, 0.6 pmol oligo(dT)18 (Thermo Fisher), 8 pmol random hexamers (TaKaRa), and 1.5× PrimeScript enzyme mix in RNase-free water) or 2 μL of RT-RamDA mix (1.5× PrimeScript buffer, 0.6 pmol oligo(dT)18, 8 pmol random hexamers or NSRs, 0.2 U of DNase I Amplification Grade (Thermo Fisher), 100 ng of T4 gene 32 protein (Roche), and 1.5× PrimeScript enzyme mix in RNase-free water) was added to 1 μL of diluted, denatured template RNA. The mixture was agitated for 30 s at 2000 rpm using MixMate (Eppendorf) and incubated in a C1000 thermal cycler (Bio-Rad) at 25 °C for 10 min, 30 °C for 10 min, 37 °C for 30 min, 50 °C for 5 min, and 85 °C for 5 min. The RT product was diluted 1:9 in nuclease-free water and used for qPCR analysis.

**Preparation of poly(A) RNA-depleted RNA and poly(A) RNA**. To evaluate triplicate samples, template RNAs were extracted from three different culture dishes for each ES and PrE condition. We prepared both poly(A)-depleted RNA and poly(A)-selected RNA from each 1 μg of total RNA using the NEBNext Poly (A) mRNA Magnetic Isolation Module (NEB) as follows. First, poly(A) RNA was bound to magnetic oligo-dT beads. We collected magnetic beads and supernatant for subsequent purification. Magnetic beads, which bound to the poly(A) RNA, were processed in accordance with the manufacturer's protocol. Finally, we obtained purified poly(A)-selected RNA. Purified poly(A)-depleted RNA was also obtained from the supernatant using RNA Clean & Concentrator-5 (Zymo Research).

**RT-qPCR for classifying non-poly(A) and poly(A) RNA**. We performed RT-qPCR using the abovementioned total RNA, poly(A)-depleted RNA, and poly(A)-selected RNA. The input amount was adjusted for the total RNA derived from 20 ng. One microliter of diluted RNA was added to 2 μL of RT mix (1.5× VILO Reaction Mix (Thermo Fisher) and 1.5× SuperScript Enzyme Mix (Thermo Fisher) in RNase-free water). RT was conducted at 25 °C for 10 min, 42 °C for 60 min, and 85 °C for 5 min. The RT product was diluted 1:25 for qPCR analysis.

**Single-cell RT-qPCR**. The single-cell lysate was thawed at 4 °C and centrifuged. Next, 0.5 μL of genomic DNA digestion mix (0.1 U of DNase I Amplification Grade (Thermo Fisher) and 2× DNase I Reaction Buffer (Thermo Fisher) in RNase-free water) was added to 1 μL of the single-cell lysate in a 96-well PCR plate and incubated at 25 °C for 5 min. After genomic DNA digestion, we added 0.5 μL of denaturing mix (8 mM EDTA and 0.02% NP40 in RNase-free water) to the digested sample, followed by incubation at 70 °C for 5 min to inactivate DNase I and desaturate the RNAs. The sample plate was immediately placed on ice. One microliter of the RT mix (3× VILO Reaction Mix (Thermo Fisher) and 3× SuperScript Enzyme Mix (Thermo Fisher) in RNase-free water) was added to the sample plate and incubated at 25 °C for 10 min, 42 °C for 60 min, and 85 °C for 5 min. For single-cell RT-qPCR in Supplementary Fig. 2d, the RT product was diluted 1:5 in RNase-free water for qPCR analysis. For scRT-qPCR in Fig. 3d, e, Supplementary Fig. 11f, and Supplementary Fig. 15g, we prepared 10× pooled primer mix containing 500 nM each of the gene-specific primers listed in Supplementary Data 2. Seventeen microliters of the preamplification mix (10 μL of 2× Platinum Multiplex PCR Master Mix (Thermo Fisher), and 2 μL of 10× pooled primer mix in nuclease-free water) was added to 3 μL of the RT products. The PCR conditions were as follows: activation at 95 °C for 2 min; 14 cycles of denaturation at 95 °C for 30 s; annealing at 60 °C for 90 s; and extension at 72 °C for 60 s. After PCR preamplification, the PCR products were added 8 μL of primer digestion mix (32 U of Exonuclease I (NEB) and 1× Exonuclease I Reaction Buffer (NEB) in nuclease-free water) and incubated at 37 °C for 30 min and 80 °C for 15 min. The final products were diluted 1:87.6 and 1.5 μL of the diluted products were used for qPCR analysis. The copy number of gene expression was adjusted and estimated using the input copy number of the spike RNAs (Lys: 1000 copies; Thr: 5 copies).

**Quantitative PCR**. qPCR was performed using a LightCycler 480 (Roche) under the following conditions: 3.5 μL of qPCR reaction mix (2.5 μL of 2× QuantiTect SYBR Green Master Mix; 0.03 μL of 100 μM forward primer; 0.03 μL of 100 μM reverse primer; and 0.94 μL of 0.0015% NP40) was added to 1.5 μL of diluted cDNA using a 384-well transfer plate (Watoson). The PCR conditions were as follows: activation at 95 °C for 15 min; 40 cycles of denaturation at 95 °C for 15 s; and extension at 60 °C for 1 min. A melting curve analysis was performed by cycling at 95 °C for 15 s, 60 °C for 15 s, and 95 °C for 15 s. The standard curve for absolute values was generated using a fivefold dilution series of Lys and Thr DNA mix as the standard (15 625, 3125, 626, 125, 25, 5, and 0 copies) as well as a fivefold dilution series of mouse genomic DNA (Clontech). Because cDNA is single-stranded, quantitative values were calculated by doubling the measured amount (dsDNA copies). Supplementary Data 2 show the primer sequences. Data analysis was performed using LightCycler 480 software, version 1.5 (Roche).

**Not-so-random primers**. Mouse first-strand NSRs consisting of 408 hexamers were synthesized individually by Sigma-Aldrich as previously reported[22]. In contrast, second-strand NSRs were designed and synthesized as a sequence complementary to the first-strand NSRs according to a previous study[21]. For details regarding the NSR sequences, please refer to Supplementary Data 3.

**Sequencing library preparation for RamDA-seq**. For the 10 pg of total RNA sample, 10 pg of total RNA was diluted in cell lysis buffer from 100 ng/μL frozen stock of total RNA. For the single-cell lysate sample, −80 °C stocks of the single-cell lysates in 96-well PCR plates were thawed at 4 °C and centrifuged. The cell lysates or 10 pg of total RNA was denatured at 70 °C for 90 s and held at 4 °C until the next step. To eliminate genomic DNA contamination, 1 μL of genomic DNA digestion mix (0.5× PrimeScript Buffer, 0.2 U of DNase I Amplification Grade, 1: 5 000 000 ERCC RNA Spike-In Mix I (Thermo Fisher) in RNase-free water) was added to 1 μL of the denatured sample. The mixtures were agitated for 30 s at 2000 rpm using an ThermoMixer C at 4 °C, incubated in a C1000 thermal cycler at 30 °C for 5 min and held at 4 °C until the next step. One microliter of RT-RamDA mix (2.5× PrimeScript Buffer, 0.6 pmol oligo(dT)18, 8 pmol 1st-NSRs, 100 ng of T4 gene 32 protein, and 3× PrimeScript enzyme mix in RNase-free water) was added to 2 μL of the digested lysates. The mixtures were agitated for 30 s at 2000 rpm and 4 °C, and incubated at 25 °C for 10 min, 30 °C for 10 min, 37 °C for 60 min, 50 °C for 5 min, and 94 °C for 5 min. Then, the mixtures were held at 4 °C until the next step. After RT, the samples were added to 2 μL of second-strand synthesis mix (2.25× NEB buffer 2 (NEB), 0.625 mM each dNTP Mixture (TaKaRa), 40 pmol 2nd-NSRs, and 0.75 U of Klenow Fragment (3′ → 5′ exo-) (NEB) in RNase-free water). The mixtures were agitated for 30 s at 2000 rpm and 4 °C, and incubated at temperatures increasing from 4 to 37 °C at a rate of 1 °C/min. Subsequently, the mixtures were maintained at 37 °C for 30 min and then at 4 °C until the next step. Sequencing library DNA preparation was performed using the Tn5 tagmentation-based method with 2/5 volumes of the Nextera XT DNA Library Preparation Kit (Illumina) according to the manufacturer's protocol. The above-described double-stranded cDNAs were purified by using 15 μL of AMPure XP SPRI beads (Beckman Coulter) and a handmade 96-well magnetic stand for low volumes. Washed AMPure XP beads attached to double-stranded cDNAs were directly eluted using 6 μL of 1× Tagment DNA Buffer (Illumina) and mixed well using a vortex mixer and pipetting. Thirteen cycles of PCR were applied for the library DNA. After PCR, sequencing library DNA was purified using 1.2× the volume of AMPure XP beads and eluted into 24 μL of TE buffer.

Advanced method for sequencing library DNA preparation for RamDA-seq: The RT-RamDA cDNA amplification performance was influenced by the quality of T4 gene 32 protein. We confirmed that T4 gene 32 protein manufactured by Roche (presently supplied by Sigma-Aldrich) was not stable, depending on its lot, for amplification performance. Therefore, we used the T4 gene 32 protein manufactured by NEB, which was more stable, and changed the incubation time at 37 °C from 60 to 30 min. During second-strand synthesis, we also confirmed that byproducts derived from the oligo-dT primers inhibited library preparation. To overcome this issue, we modified the method as follows: the concentration of NEB buffer 2 was changed from 2.25× to 2.5×, and the reaction conditions were changed to 16 °C for 60 min, 70 °C for 10 min, and maintenance at 4 °C until the next step. Sequencing library preparation using the Nextera XT DNA Library Preparation Kit was performed in a 1/4 volume using 14 PCR cycles. For analyses of batch effect to evaluate plate-to-plate variability, we prepared RamDA-seq library DNA by using this advanced method.

**Sequencing library preparation for C1-RamDA-seq**. The ES and PrE cell suspensions were adjusted to $1 \times 10^6$ cells/mL and stained with 1 μg/mL Calcein AM and Calcein Blue AM (Thermo Fisher) in PBS at 37 °C for 5 min, respectively. After the cell suspensions were stained, they were combined at a ratio of 1:1 and diluted in PBS to $3 \times 10^5$ cells/mL. Sixty microliters of the diluted cell suspension was mixed with 40 μL of C1 Suspension Reagent (Fluidigm). Six microliters of this Cell Mix was loaded into the new designed C1 Single-Cell Open App IFC 1862× (cells 10–17 μm in diameter). The captured cells were stained with 2 μg/mL PI solution using IFC to identify dead cells. Bright-field and fluorescence imaging of every capture site was performed using an Olympus IX83 microscope system with

MetaMorph software (Molecular Devices). In addition, we carefully defined ES-single, PrE-single, doublet, dead-cell, and not-captured sites. The C1-RamDA-seq script was created using Script Builder software 2.1.1. (Fluidigm). Each reaction component for C1-RamDA-seq was as follows: Lysis Final Mix (1.12 μL of 10% NP40, 4.05 μL RealTime ready Cell Lysis Buffer, 0.84 μL of 40 U/μL RNasin Plus RNase Inhibitor, 3 μL of 1:5000 ERCC RNA Spikes, 1.35 μL of C1 Loading Reagent, and 16.55 μL of RNase-free water), gDNA Digestion Final Mix (2.5 μL of 5× PrimeScript Buffer, 5 μL of 1 U/μL DNase I Amplification Grade, 1 μL of 20× C1 Loading Reagent, and 11.5 μL of RNase-free water), Priming Final Mix (17.23 μL of 5× PrimeScript Buffer, 5.24 μL of PrimeScript RT Enzyme Mix, 0.7 μL of 30 μM oligo(dT)12, 2.8 μL of 100 μM 1st-NSRs, 1.75 μL of 2 mg/mL T4 gene 32 protein (NEB), 1.13 μL of C1 Loading Reagent, and 1.15 μL of RNase-free water), RT Final Mix (12 μL of 5× PrimeScript Buffer, 3 μL of PrimeScript RT Enzyme Mix, Real-Time ready Cell Lysis Buffer, 0.4 μL of 30 μM oligo(dT)12, 1.6 μL of 100 μM 1st-NSRs, 1 μL of 2 mg/ml T4 gene 32 protein (NEB), 3.96 μL of 1 U/μL DNase I Amplification Grade, 2.25 μL of C1 Loading Reagent, and 33.92 μL of RNase-free water), Second-strand Final Mix (6.7 μL of 10× NEB buffer 2, 6.7 μL of 2.5 mM each dNTP Mixture, 5.36 μL of 100 μM 2nd-NSRs, 2.01 μL of 5 U/μL Klenow Fragment (3′ → 5′ exo-), 1.5 μL of C1 Loading Reagent, and 7.73 μL of RNase-free water), and Harvest Reagent (500 μL of Tagment DNA Buffer, 237.5 μL of C1 Harvest Reagent, and 12.5 μL of 20× C1 Loading Reagent).

For the 10-pg total RNA sample, 30 ng of total RNA was added to the Lysis Final Mix, and the Cell Wash Buffer (Fluidigm) was loaded for IFC rather than Cell Mix. The thermal conditions were as follows: lysis step (chamber 1: 4 °C for 1 s, 70 °C for 90 s, and 4 °C for 300 s); gDNA digestion step (chambers 1–2: 130 °C for 300 s and 4 °C for 1 s); priming step (chambers 1–3: 25 °C for 600 s and 30 °C for 600 s); RT-RamDA step (chambers 1–4: 37 °C for 900 s, 50 °C for 300 s, 94 °C for 300 s and 4 °C for 1 s); and second-strand synthesis step (chambers 1–5: 16 °C for 3600 s and 75 °C for 1200 s). We recovered 3 μL of C1 products from IFC and directly added 1 μL of Amplicon Tagment Mix to a 1/5 volume of the Nextera XT DNA Library Preparation kit. We performed 14 cycles of PCR to evaluate the library DNA in this kit. After PCR enrichment, sequencing library DNA was purified in 1.2× the volume of AMPure XP beads and eluted into 24 μL of TE buffer. The typical yield of the library DNA was ~25 ng. The average length of library DNA was ~300 bp.

**Sequencing library preparation for SMART-Seq v4**. Amplified cDNA from 10 pg of total RNA was prepared using the SMART-Seq v4 Ultra Low Input RNA Kit for Sequencing (Clontech) according to the manual instructions. One microliter of 10 pg/μL total RNA with 1:5 000 000 ERCC RNA Spike-In Mix I in RNase-free water was added to 10.5 μL of the reaction buffer (1 μL of 10× reaction buffer in RNase-free water). Sequentially, we added 1 μL of 3′ SMART-Seq CDS Primer II A (12 μM) to the sample before the denaturation step. We performed 17 cycles of PCR for cDNA amplification. The amplified cDNA was purified using AMPure XP beads and eluted with 17 μL of TE buffer. The cDNA yield and average length of amplified cDNA were calculated using the Bioanalyzer Agilent High-Sensitivity DNA Kit (Agilent Technologies) in the range of 400–10 000 bp. Library DNA was prepared using 62.5 pg of amplified cDNA for a 1/4 volume of the Nextera XT DNA Library Preparation Kit according to the manufacturer's protocol. Using this kit, we performed 12 cycles of PCR for the library DNA. For evaluation of reproducibility in Supplementary Fig. 4f, we carried out SMART-Seq v4 library preparation using Fluidigm C1. The script "Full-length mRNA Sequencing" was downloaded from Script Hub (https://jp.fluidigm.com/c1openapp/scripthub), and we prepared C1-SMART-Seq v4 library DNA according to the manual instructions. To adjust loading amount of total RNA as 10 pg per IFC chamber, we prepared the 20 μL of Lysis Mix (2.4 μL of 3′ SMART-Seq CDS Primer II A (12 μM), 2 μL ES total RNA (11.11 ng/μL), 2 μL of 1:4500 ERCC RNA Spikes, 2.6 μL of 10× Reaction Buffer, 1 μL of C1 Loading Reagent, and 10 μL of RNase-free water).

**Sequencing library information of RamDA-seq**. When we performed quality control of RamDA-seq library DNA using the Bioanalyzer Agilent High-Sensitivity DNA Kit, the typical yield of the sequencing library DNA obtained from one mESC or 10 pg of total RNA was 120–150 ng. The length was ~300 bp. To investigate the possibility of generating library DNAs derived from genomic DNA or environmental DNAs, we prepared transcriptase-minus and non-template control samples. Thus, we confirmed that the RamDA-seq protocol accompanied by DNase I digestion did not produce library DNA from genomic DNA and environmental DNAs (Supplementary Fig. 3).

**Quality control and sequencing of library DNA**. All of the samples prepared with Nextera XT DNA Library Preparation (including RamDA-seq, C1-RamDA-seq, SMART-Seq v4, and C1-SMART-Seq v4) were quantified and evaluated using a MultiNA DNA-12000 kit (Shimadzu) with a modified sample mixing ratio (1:1:1; sample, marker, and nuclease-free water) in a total of 6 μL. The length and yield of the library DNA were calculated in the range of 150–3000 bp. The library DNA yield in particular was estimated as 0.5 times the value quantified from the modified MultiNA condition. Subsequently, we pooled each 200 fmol of library DNA in each well of a 96-well plate. The pooled library DNA was evaluated based on the averaged length and concentration using a Bioanalyzer Agilent High-Sensitivity

DNA Kit in the range of 150–3000 bp and a KAPA library quantification kit (Kapa Biosystems). Finally, 1.5 pM pooled library DNA was sequenced using Illumina NextSeq 500 (single-read 76 cycle sequencing).

For rdRNA-seq and paRNA-seq using 1 μg total RNA, library DNAs were quality controlled by a Bioanalyzer Agilent High-Sensitivity DNA Kit and a KAPA library quantification kit and sequenced using Illumina HiSeq 2500 (paired-end read 101 cycle sequencing). For rdRNA-seq using 10 and 1 ng total RNA, library DNA was sequenced using Illumina NextSeq 500 (single-read 76 cycle sequencing).

**Analyses of single-cell preamplified RT-qPCR data.** Copy numbers were log10-transformed using a pseudocount of 1. For subsequent analyses, we removed cells with a copy number of one for the housekeeping genes (*Gnb2l1* and *Eef1b2*) lower than $Q1 - 1.5$ interquartile range (IQR). To compare scRT-qPCR data with RamDA-seq data with ES cells sorted by cell cycle phases, we classified ES cells at 0 h for scRT-qPCR data into G1, S, and G2M according to the DNA abundance quantified using Hoechst 33342 (355–448/59–Area) as follows: G1 if $x < 24\,000$, G2M if $x > 38\,000$; otherwise S. To compare scRT-qPCR data with RamDA-seq data with cells sampled across the ES-PrE time series, we constructed a diffusion map of scRT-qPCR data using the "destiny" R package and used DC1 as the pseudotime.

**Analysis of exonic regions covered by the sequenced reads.** For each scRNA-seq method, the FASTQ files of sequencing data with 10 pg of RNA were combined. Fastq-mcf (version 1.04.807)[47] was used to trim adapter sequences and generate read lengths of 42 nucleotides (nt) with the parameters "-L 42 -l 42 -k 4 -q 30 -S." Seqtk (version 1.1-r93-dirty; https://github.com/lh3/seqtk) was used to downsample the reads to the smallest number of reads among all methods (46 600 826). For comparison, we also prepared R1 reads of rdRNA-seq and paRNA-seq data. The reads were mapped to the mouse genome (mm10) using HISAT2 (version 2.0.1)[48] with parameters "--dta-cufflinks -p 4 -k 5 --sp 1000,1000." Uniquely mapped reads were selected using the BAMtools (version 2.0.1)[49] "filter" command with the parameters "-isMapped true -tag NH:1" and the SAMTools (version 2.0.1)[50] "view" command with the parameter "-q 40." BEDTools (version 2.22.1)[51] and R were used to calculate the fraction of exonic regions covered by the sequenced reads. A base was defined as covered if at least one read overlapped the base.

**Histone-coding gene analysis.** GENCODE transcripts with gene names starting with "Hist" and transcript types of "protein_coding" were selected as histone-coding genes. Heat maps were generated using "aheatmap" function in the "NMF" R package[52]. TPM values for each transcript were quantified using the sailfish (version 0.9.2)[53] "quant" command with the parameter "-l U."

**Quantification of expression levels of transcripts and ERCC.** Transcript-level expression levels were quantified in the unit of count or TPM using the sailfish "quant" command with the parameter "-l U." A FASTA file consists of sequences of ERCC RNAs and transcripts of the GENCODE vM9 annotation used as reference for sailfish.

**Transcriptome assembly and non-poly(A) RNA identification.** Fastq-mcf was used to trim adapter sequences with the parameters "-l 50 --lowcomplex-pct 36 --homopolymer-pct 36 -k 4 -S." The reads were mapped to the mouse genome (mm10) using HISAT2 with the parameters "--dta-cufflinks --rna-strandness RF -k 5 --no-mixed --no-discordant --sp 1000,1000." Properly (i.e., convergent read pairs) and uniquely mapped reads were selected using the BAMtools "filter" command and SAMTools "view" command with the parameter "-q 40." For genome-guided transcriptome assembly, Cufflinks was used with the parameters "--multi-read-correct --frag-bias-correct -M $mask --library-type fr-unstranded." A GTF file of tRNA and rRNA annotations in GENCODE (vM9) was provided to mask the genome. Cuffcompare was used to annotate transcripts with transfrag class codes with respect to the GENCODE (vM9) annotation (Cufflinks website). According to the Cuffcompare class codes, we selected (1) unannotated transcripts with class codes of "i", "o", "u", "x", or "s" and (2) unannotated splicing variant transcripts ("j" class (potentially novel isoform)) with at least one unannotated splice junction. We then removed (1) unannotated transcripts with exons located within 100 bp of the tRNA or rRNA annotations in GECODE or RepeatMasker or pseudogene annotations in GENCODE (2wayconspseudos) and (2) unannotated transcripts with lengths that were not longer than 200 bp. These filtered unannotated gene models were further merged with gene models in GENCODE vM9.

Using the merged gene models, the expression levels of bulk total and poly(A) RNA-seq data were quantified using the sailfish "quant" command with the parameter "-l ISR." The resulting "NumReads" data were used for differential expression analyses between total and poly(A) RNA-seq in ES or PrE using the EdgeR (version 3.12.1)[54] "glmTreat" function with the parameter "lfc = 0.5." Non-poly(A) transcripts were called when, for either ES or PrE samples, the following criteria (loose criteria) were obtained: FDR < 0.05 and averaged fitted values in the total RNA-seq of at least 10. For non-poly(A) transcripts with strict criteria, an averaged fitted value in the poly(A) RNA-seq <1 was also required. Poly(A) transcripts were called similarly as were loose criteria. We further removed

unannotated transcripts with a class code of "i" from the non-poly(A) transcript definition because they were difficult to distinguish from pre-mRNA or spliced intronic fragments. Note that *Neat1-001* was not included in the above non-poly (A) transcript definition potentially because of its polyadenylated isoform (*Neat1-002*), which was completely included in the gene body of *Neat-001*. This exclusion resulted in the inaccurate expression quantification of the two transcripts for total and poly(A) RNA-seq data. ES-expressed non-poly(A) RNAs were defined as transcripts with averaged fitted values in the total RNA-seq of at least 10 for the ES samples. The ES-enriched, PrE-enriched, and unchanged subsets of non-poly (A) RNAs were defined according to the differential expression pattern between ES and PrE. Lists of non-poly(A) and poly(A) transcripts are shown in Supplementary Data 4 and 5, respectively.

**Evaluation of performance for detecting non-poly(A) RNAs.** To evaluate the performance for detecting non-poly(A) transcripts for each scRNA-seq method, sequenced reads with 10 pg of RNA for each scRNA-seq method were trimmed as described above and downsampled to the lowest number of reads in the data set (7 489 702) using seqtk. The expression level was quantified using the sailfish "quant" command with the parameter "-l U" on the above merged gene models. We selected non-poly(A) transcripts with averaged fitted values of at least 10 in ES as ES-expressed non-poly(A) transcripts. A non-poly(A) transcript was identified if the TPM was at least 0.1. The TPM values were transformed to logarithm of base 10 with a pseudocount of 1. For ES-expressed non-poly(A) transcripts with averaged TPM values of at least 0.1, a Pearson correlation coefficient was calculated between the TPM of the total RNA-seq and each scRNA-seq method. Analyses using RamDA-seq with cells were performed as described for data with 10 pg of RNA.

**Quality assessment of RamDA-seq and C1-RamDA-seq with cells.** Preprocessing: The 'no Mix' samples, which showed no amplification, were removed from the following analyses. Fastq-mcf was used for adapter trimming with the parameters "-l 36 --lowcomplex-pct 74 --homopolymer- pct 74 -k 4 -S all_se-quencing_WTA_adopters.fa" for RamDA-seq and "-L 75 -l 46 -k 4 -q 30 -S" for C1-RamDA-seq.

Quantification of a proportion of rRNA reads in FASTQ: The trimmed reads were mapped to (mouse) rRNA sequences (Supplementary Data 6) using HISAT2 with the parameters "--dta-cufflinks -p 4 -k 5 -X 800 --sp 1000,1000." The proportion of rRNA reads in FASTQ was calculated by dividing the sum of reads "aligned exactly 1 time" and "aligned >1 times" reported by HISAT2 by the total number of reads.

Genome mapping: The reads were mapped to the mouse genome (mm10) using HISAT2 with the parameters "--dta-cufflinks -p 4 -k 5 -X 800 --sp 1000,1000." Uniquely mapped reads were selected using the BAMtools "filter" command with the parameters "-isMapped true -tag NH:1" and the SAMTools "view" command with the parameter "-q 40."

Expression level quantification: Transcript-level expression levels were quantified in the unit TPM using the sailfish (version 0.9.2)[53] "quant" command with the parameter "-l U." The GENCODE vM9 annotation and merged gene models (described above) were used.

Number of detected transcripts: Transcript-level expression levels quantified using the GENCODE vM9 annotation were used to calculate the number of detected transcripts.

PCA: For C1-RamDA-seq, PCA was performed with log-transformed TPM data quantified by sailfish using "prcomp" in R.

Defining "outlier cells": We defined "outlier cells" as follows. The numbers of reads mapped to rRNA annotations in GENCODE vM9 were counted using the featureCounts program in Subread. Next, we defined outlier cells as cells for which (1) the number of uniquely mapped reads was lower than $Q1 - 1.5 \times IQR$ or (2) the ratio of rRNA reads was greater than $Q3 + 1.5 \times IQR$ (Supplementary Fig. 15e). We removed these outlier cells from subsequent analyses.

**Analyses of cell cycle data.** The transcript-level expression levels of the merged gene models (described above) were used. After removing five samples (2 for G1, 2 for S, and 1 for G2M) as outliers in the primary diffusion map analysis, we performed a diffusion map analysis as follows. First, we selected expressed transcripts with a TPM of at least 1 in at least 10% of cells. For each transcript among the expressed genes, we first calculated averaged TPM across cells in each cell cycle phase and then calculated the CV using the averaged TPM values for G1, S, and G2M. For each cell cycle phase, we selected the top 5000 high-CV transcripts with an averaged TPM that was higher than those for the other phases. Using the selected transcripts, we performed diffusion map analysis using the "destiny" R package (version 1.0.0)[55]. We used the DC1 as the pseudotime, i.e., we treated the expression levels as a function of DC1. We fitted the sine function to the pseudotime-series data for each transcript using the "lm" function in R. The FDRs were calculated from the p-values for multiple testing corrections using the Benjamini and Hochberg procedure. The transcripts with an FDR < 0.01 were called oscillating transcripts. For visualization, the raw values were smoothed by fitting the sine function. Heat maps were generated using "aheatmap" function in the "NMF" R package (version 0.20.6).

We also conducted an "unsupervised" selection of highly variable genes. For each of the expressed transcript with a TPM of at least 10 in at least 10% of cells, we calculated the mean and CV across G1, S, and G2M cells (FDR < 0.01). We then fitted the mean-$CV^2$ relationship and searched for genes with significant deviation from the fit (as described in http://pklab.med.harvard.edu/scw2014/subpop_tutorial.html) (p-value adjusted using the Benjamini–Hochberg procedure <0.01). Using the selected transcripts, we performed diffusion map analysis using the "destiny" R package.

### Analyses of time-series data.

The transcript-level expression levels of the merged gene models (described above) were used. We conducted a diffusion map analysis as follows: (1) we selected expressed transcripts with a TPM of at least 10 in at least 10% of cells. (2) We calculated the CV of TPM for each of the expressed transcripts and then selected the top 5000 high-CV transcripts. (3) We performed diffusion map analysis on the expression data of the selected transcripts using the "destiny" R package. (4) We used DC1 as the pseudotime, i.e., we treated the expression levels as a function of DC1. We fitted a generalized additive model (GAM) to the log-transformed pseudotime-series data for each transcript using the "mgcv" R package (version 1.8–16) with the parameter "family = Gaussian(link = identity)." The FDRs were calculated using the p-values for multiple testing corrections according to the Benjamini and Hochberg procedure. The Akaike information criterion (AIC) was calculated for GAM and an intercept model. The transcripts with an FDR < 0.01 and an AIC that was greater for GAM than for the intercept model were called dynamically regulated transcripts. Hierarchical clustering was performed using the "flashClust" R package (version 1.01–2) according to Ward's method and 1 – Pearson correlation coefficient as the distance. For interpretability, we clustered dynamically regulated transcripts into seven clusters. Heat maps were generated using "aheatmap" function in the "NMF" R package (version 0.20.6). For visualization, the raw values were smoothed by fitting the GAM. Functional enrichment analyses were performed for each cluster using Metascape (http://metascape.org)[56]. Heat map representations of the genomic coverage of the RamDA-seq data were generated using Millefy (https://github.com/yuifu/millefy).

### Analysis of recursive splicing.

For the analysis of RS, we used the genome mapping data (as described above). We first selected RS-site candidates using novel splice junctions detected by RamDA-seq with similar criteria as in a previous study[13]: (1) long (>150 kb) introns not overlapped with exons of any other transcripts; (2) novel junctions with one anchor sequence corresponding to a known splicing boundary ("partially novel" by RSeQC); (3) anchor sequences corresponding to intronic regions have pentamers found at 1% of all 5′ splice sites (GTAAG, GTGAG, GTAGG, GTATG, GTAAA, GTAAT, GTGGG, GTAAC, GTCAG, GTACG, GTACA, GTATT, GTACT, GTGTG, GTGCG, and GTACC) for 5′ splice sites or 3′ splice motif (polypyrimidine tract consisting of >11 pyrimidines present in the region of −22 to −1, including YAG as last three positions) for 3′ splice sites; (4) split alignment reads have >10-nt overhang; and (5) >5-kb junction regions. These criteria yielded 207 RS-site candidates in 77 genes.

Next, we searched for sawtooth wave patterns in the >150-kb introns, similar to a previous study[13]. We normalized the read coverage for each cell in the differentiation time course data by the number of mapped reads and then summed the normalized read coverage across all cells. The summed read coverage was partitioned into 5-kb bins. We coded the position of the RS-site candidates using a binary dummy variable indicating whether the positions were between the upstream exons or the RS-site candidates. We then fitted two linear regression models to the read coverage: one with genomic position as an explanatory variable (the baseline model) and the other with genomic position and the dummy variable as explanatory variables (the augmented model). We used an F-test p-value (p < 1e-5) to quantify the improvement of the goodness of fit provided by each RS-site candidate. We further asked whether the fitted augmented model showed a sawtooth wave pattern using the intercept, slope coefficient, RS-site coefficient, and augmented/baseline slope ratio of the fitted models. The above filtering steps retained three genes: Cadm1, Robo2, and Magi1.

Based on a simulation, we estimated the sensitivity of RS detection as a function of the number of reads mapped to the intronic regions with RS-site candidates. For Cadm1, Robo2, and Magi1, intronic read coverage was aggregated across cells at a time point when host gene expression was highest (00 h for Cadm1 and Magi1; and 72 h for Robo2). Then, the aggregated intronic read coverage data were repeatedly subsampled (100 times for each subsampling fraction: $1/10^{3.0}$; $1/10^{2.5}$; $1/10^{2.0}$; $1/10^{1.5}$; $1/10^{1.0}$; $1/10^{0.5}$; and $1/10^{0.0}$. For each subsampling fraction, linear regression models were fitted against the read coverage replicates to detect RS (p < 1e-5). Note that the proportion of trials where RS was detected can be interpreted as RS detection probability. Finally, RS detection probability as a function of the number of intronic reads was estimated by logistic regression. For interpretability, we converted the number of intronic reads into gene-level TPM of host genes calculated from the aggregated gene-level TPM across cells from the corresponding time point.

We searched for RS in each single cell as follows. Using the estimated sensitivity of RS detection above, we selected cells with sufficient intronic reads (RS detection probability >0.95 or >0.99). Additionally, only cells with reads within intronic regions both upstream and downstream of candidate RS sites were selected. For the selected cells, the linear regression models were fitted against the intronic read coverage as described above.

The above analysis was performed using custom R and Julia scripts and the Bio.jl (https://github.com/BioJulia/Bio.jl) package.

### Enhancer RNA analysis.

For the analysis of enhancer RNA, we used genome mapping data (as described above).

The permissive set of mouse enhancers identified using FANTOM5 CAGE data was downloaded from the FANTOM website (http://fantom.gsc.riken.jp/5/datafiles/latest/extra/Enhancers/). The genomic coordinates of the enhancer annotation were converted from mm9 to mm10 using liftover (Kent et al., 2002). Enhancers located within 2 kb of the GENCODE (vM9) annotation were removed. We also removed enhancers wider than 400 bp to avoid complications while merging the enhancer regions. The number of reads overlapping the ±400-bp region in the center of each enhancer (801 bp) was counted using the featureCounts program in Subread (version 1.5.1)[57] with parameter "-O" and normalized to the total number of mapped reads. Using the filtered enhancer set, we defined positive and negative control sets of enhancers using CAGE TPM in mESC samples (ES-OS25 embryonic stem cells, untreated control). A positive control set of enhancers was defined as enhancers with an averaged CAGE TPM above the 90% quantile among the filtered enhancer set. The negative control set of enhancers was defined as enhancers with averaged normalized total and poly(A) counts of 0 and an averaged CAGE TPM of 0. The normalized read counts for the 10 pg of RNA data for each scRNA-seq method were calculated as described above. An eRNA was defined as detected if the normalized read counts exceeded 0.1. The read coverage matrices around enhancers were computed using deepTools (version 2.2.4)[58]. Aggregation plots and heat maps were generated using custom R scripts.

Motif enrichment analyses were performed using the "findMotifsGenome" function in HOMER[59] with the parameters "-size 200 -mask." For motif enrichment analyses of eRNAs expressed in ESCs, eRNAs whose expression levels were above 0.1 in >10% of cells at 0 h were selected using RamDA-seq, and eRNAs whose expression levels were above 0.1 in at least one ESC sample were selected using bulk total RNA-seq. Only the results regarding enrichment of known motifs were used.

We defined another set of enhancers using non-poly(A) transcripts and enhancer-like histone modifications as previously described[60]. H3K4me1 and H3K4me3 ChIP-Seq data for mouse (129/Ola) ES-E14 stem cells were downloaded from the ENCODE project[61] in BAM format. The genomic coordinates of ChIP-Seq were converted from mm9 to mm10 using liftover. The number of reads mapped within ±500 bp of the TSS of each transcript was counted and normalized to the total number of mapped reads. The normalized counts were used to calculate log2(H3K4me1/H3K4me3) values. We defined non-poly(A) transcripts with log2(H3K4me1/H3K4me3) values >0.58 as eRNAs. The eRNAs were detected as described above for the non-poly(A) transcripts. The cells were analyzed using RamDA-seq as described for the 10-pg RNA samples.

### Code availability.

All custom computer codes in the generation or processing of the described data are available at https://github.com/yuifu/Hayashi2018.

### Data availability.

Data generated in the study can be accessed at the Gene Espression Omnibus under accession code GSE98664. Previously published data sets used in this study can be accessed from the Gene Expression Ombinus as follows:

Quartz-Seq: GSE42268, SUPeR-seq: GSE53386.

All other data are available from the authors upon reasonable request.

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

## Acknowledgements

We thank the members of the Bioinformatics Research Unit, particularly Masashi Ebisawa for assistance of experiments, Koki Tsuyuzaki and Mika Yoshimura for discussion of data analyses, and Manabu Ishii and Akihiro Matsushima for the management of IT infrastructure. We also thank Hitoshi Niwa (Laboratory for Stem Cell Biology, Institute of Molecular Embryology and Genetics at Kumamoto University) for providing the ES and PrE cells. This work was partially supported by the Platform Project for Supporting Drug Discovery and Life Science Research (Platform for Drug Discovery, Informatics, and Structural Life Science) by Innovative Technology from the Ministry of Education, Culture, Sports, Science and Technology (MEXT) of Japan and Japan Agency for Medical Research and development (AMED). This study was also partially supported by the Projects for Technological Development, Research Center Network for Realization of Regenerative Medicine by AMED and Japan Science and Technology Agency (JST). This work was supported by JSPS KAKENHI Grant Numbers JP24651218, JP26640120, and MEXT KAKENHI (No. 221S0002). The Special Postdoctoral Researchers Program from RIKEN also supported this work. This work was also supported by JST CREST Grant Number JPMJCR16G3, Japan.

## Author contributions

T.H., Y.S., and I.N. designed and configured the various approaches used in this study. T.H. and Y.S. performed the majority of the experiments. H.O., T.H., and I.N. analyzed the data and developed the bioinformatics methods and tools. T.H. performed the cell sorting and assisted with the single-cell qPCR. H.D. assisted with the cellular experiments. M.U. assisted with the RamDA-seq experiments. T.H., H.O., and I.N. prepared the figures and wrote the manuscript. All authors read and approved the final manuscript.

## Additional information

**Competing interests:** T.H., Y.S., and I.N. are inventors on a patent application related to this RT and global cDNA amplification procedure. The remaining authors declare no competing financial interests.

