## [Peer Review File · Nature Communications]

Reviewers' comments:

Reviewer #1 (Remarks to the Author):

In this manuscript, Hayashi and colleagues describe a novel total RNA approach for single-cell RNA sequencing. They demonstrate that it successfully captures both poly-A and non-poly-A transcripts, with low noise and consistent coverage across the transcript. They also apply their method to uncover some interesting single-cell biology related to splicing and enhancer activity. I found the manuscript to be interesting and well-written, and the benefits of the proposed approach are clear from a technical and biological perspective. However, I have some concerns relating to the performance of the technology and the computational data analysis - see my comments below for details. If these are addressed, I see no reason why this work would not be a useful contribution to the single-cell transcriptomics field.

MAJOR:

1. The authors use NSR primers to reduce the representation of rRNAs that would otherwise dominate the sequencing output. This seems to be a critical step for the protocol to be useful - however, I assume that this process is not perfect. What is the percentage of reads in the sequencing libraries that correspond to ribosomal RNA? How does this compare to ribosomal depletion strategies involving probes for capture or degradation? (Note that quantification from sequencing data requires some care, as the many rRNA copies may preclude alignment and counting if done naively.)
2. Some more elaboration on the sample sizes would be desirable. How many diluted mESC RNA samples were generated with each protocol to calculate the CV2 values in Figure 1E? I would expect around 100 samples at least, given that real plate-based scRNA-seq experiments would be expected to use the entire plate. A related concern is the number of plates/batches that were used, not just the number of wells on each plate. scRNA-seq data is subject to considerable plate effects (see <https://doi.org/10.1101/025528>, <https://doi.org/10.1038/srep39921>), and I would strongly recommend examining not just the well-to-well variability, but also the plate-to-plate variability, e.g., when plates are processed on different days, with different operators, etc.
3. With respect to the conclusion that RamDA-seq provides full-length coverage, Figure 2A may be misleading as the profile depends on the true length of transcripts. One can imagine that, even for a perfect technique that gave completely uniform coverage, the profile would not be flat if the underlying distribution of full transcript lengths was uneven. I would suggest an additional plot where the transcript lengths are scaled to a fixed interval. This ensures a protocol yielding full-length uniform coverage would also yield a flat profile.
4. The numbers on page 10 do not match up with the numbers in Figure 3B.
5. The linear regression to identify the sawtooth pattern for recursive splicing does not seem appropriate. In particular, I do not understand why the authors are using a dummy binary variable (page 52). It seems more logical to use the distance from the closest upstream RS site candidate as a real-valued covariate to model the coverage at each bin, given that coverage drops linearly with distance. A related issue is the claim that "the sawtooth wave pattern... weakened as differentiation progressed". While the decrease in coverage is clear, is this a specific reduction in recursive splicing or simply a general downregulation of the entire gene?
6. For Figure 5B, I would like to see separate coverage profiles for the forward and reverse strands. This will confirm that they are indeed eRNAs, which are characterised by inverted bimodality around the enhancer. The "background" is also constructed from random genomic regions - how were these chosen?
Were these matched for GC content, mappability/sequenceability, DNA accessibility, etc.?

Obviously, it would be trivial to obtain low "background" coverage if the background regions were defined in difficult-to-map regions such as telomeres.

7. The read processing steps are very inconsistent across the different analyses. For some analyses, alignment was performed using HISAT2 (page 42), while for others, Bowtie2 was used (page 49). For quantification of gene expression, some analyses used Sailfish (pages 42, 45), others used Cufflinks (page 43) and others used featureCounts on aligned data (page 46). It would be advisable to standardize the quantification pipeline for all analyses, to avoid confusion and simplify the interpretation of the results. The use of Cufflinks seems particularly unnecessary, as I do not see many references to newly-discovered unannotated transcripts in the manuscript.

8. The selection of genes on page 48 is not coherent. Why do the authors identify the top HVGs within each phase, and then select the genes that are DE between phases? This will artificially encourage the separation of cells from different phases on the diffusion map. If the aim is to create an unbiased diffusion map, they should just use highly variable genes detected across the entire population of cells (from all phases). Otherwise, the authors should justify their choice of genes to construct a biased pseudotime.

MINOR:

1. Genomic contamination is an important consideration for RNA-seq protocol development. There is no harm in showing the data as a Supplementary on the last line of page 6.

2. Figure 1D uses a rather unconventional metric to examine the similarity between each scRNA-seq library and the bulk data (the number of genes with counts ≥ 10 and $|FC| \leq 2$). A more natural statistic would be the Pearson's correlation between the counts, which can be directly interpreted as the proportion of variance explained. I suspect this would give similar results without the need for the arbitrary threshold on the fold-change. Otherwise, it would be necessary to show similar results at different thresholds.

3. Supplementary Figure 5C is misleading as the expression values have simply been copy-and-pasted to form a continuum. I would suggest colouring the copies differently to avoid giving the idea that there is information on the cycle number for each cell.

4. Page 49 makes references to the cell cycle phases, while the section refers to time series data. I presume that this is a mistake.

5. Consider using TREAT (specifically, the glmTreat function) to test against a log-fold change threshold, rather than post-hoc filtering on the log-fold changes on page 44. See <https://support.bioconductor.org/p/62286/#62287> for a discussion of this topic by some of the edgeR authors.

Reviewer #2 (Remarks to the Author):

In this paper Hayashi et al. propose RamDA-seq, a full length total RNA-seq method that shows near complete full-length transcript coverage, and, as the authors claim, especially high sensitivity to non-poly(A) RNA. The method was tested to reproduce dynamical trajectories of differentiating embryonic stem cells, revealing hundreds of dynamically regulated non-poly(A) transcripts. They also demonstrate that the data resulting from RamDA-seq can detect recursive splicing, enhancer RNAs and their cell-type specific activity in single-cells.

This is a well written paper, with the data well assembled and presented and the results clearly stated, and we support its publication as we think that the single-cell RNA-seq community will greatly benefit from this method. However, we have a number of major and minor comments that

we would like the authors to address before accepting the paper fully.

Major comments

1. Important details of how the RT enzyme works are missing from this paper. While we understand that it is described in greater detail in the other paper (Hayashi et al., NAR, in review) by the same authors, it would be good to add a supplementary note describing details about the underlying biology (e.g. how does strand displacement amplification work?)
2. The authors claim that "RamDA-seq protocol did not produce a library of DNA derived from genomic DNA or environmental DNAs (data not shown)". Can the authors provide some evidence of this claim? Supplementary Fig 3c shows that the proportion of intronic/intergenic reads is higher in RamDA-seq (>40%) than other methods. How do we reconcile these two statements? Some commentary is essential.
3. For Fig. 1d and 1e, I believe a useful (and easy) control would be to perform RamDA-seq on 1 microgram RNA and compare its performance to the gold standards (rdRNA-seq and paRNA-seq)
4. It is not clear if the data shown on Fig. 2b is an average across multiple single cells for RamDA-seq or coverage within a single cell. It would be useful to see what the coverage within example single cells looks like. Also it is not clear why Mdb1/Mdn1 (the name of the gene is mismatched between the figure and the text) was chosen? Can the authors comment on its significance?
5. Are non-poly(A) transcripts enriched specifically in Clusters 1,2 and 5 in Fig. 3d. If so, why?
6. Related to Fig. 3, a discussion on the biological significance of Neat1 is missing. The authors should relate to what was published
7. The authors claim that "the sawtooth wave pattern in the first intron of Cadm1 weakened as differentiation progressed, whereas the pattern in the second intron of Robo2 strengthened". But this could merely be the result of changing expression levels. In other words, the data as presented are consistent with the fact that the recursive splicing stays exactly the same, it's only that it cannot be detected when global expression levels are low. Can the authors comment on the sensitivity of RS detection as a function of expression levels?
8. Also, can the authors detect heterogeneity across single cells in the presence/absence of RS within a specific gene?
9. Lastly, but importantly, given the high proportion of rRNA, intronic and intergenic reads in RamDA-seq, can the authors comment on the depth per cell needed to get useful information out?

Minor comments

1. The often used phrase "RamDA-seq is the single-cell RNA-seq method" very strange. This does not convey any meaning and I am sure is an error. Stated this way, it appears that there is no other single-cell RNA-seq method other than RamDA-seq. I would suggest a rephrasing as follows: "Altogether, based on the substantial resemblance, we conclude that RamDA-seq is a robust single-cell total RNA-seq method"
2. For comprehensiveness, it would be good if Fig. 2b involved SUPER-seq. Also can the authors share a few more representative examples like Fig. 2b (which is impressive) in the supplement?
3. The wording of Fig. 2c within the text is not clear. What does "longer range of transcripts" mean? I guess the authors are referring to a longer range of transcripts binned by length – clarification would help.

RESPONSE TO REVIEWERS

Original reviewer comments appear below in italics with responses in blue.

RESPONSE TO REVIEWER 1:

We are thankful for the reviewer's insightful comments, which have helped us to significantly improve the paper.

Major Comments

Comment 1. The authors use NSR primers to reduce the representation of rRNAs that would otherwise dominate the sequencing output. This seems to be a critical step for the protocol to be useful - however, I assume that this process is not perfect. What is the percentage of reads in the sequencing libraries that correspond to ribosomal RNA? How does this compare to ribosomal depletion strategies involving probes for capture or degradation? (Note that quantification from sequencing data requires some care, as the many rRNA copies may preclude alignment and counting if done naively.)

Response: We appreciate your comment regarding this point. In accordance with the reviewer's comment, we evaluated the proportion of rRNA reads in the FASTQ files for scRNA-seq methods. Thus, we have added the data to the revised version of Supplementary Figure 4b and the revised version of Supplementary Figure 12c. The proportions of rRNA reads of C1-RamDA-seq and RamDA-seq were approximately 20-35% with 10 pg and approximately 10-25% with cells. Despite having a higher rRNA read ratio than rdRNA-seq, RamDA-seq showed high sensitivity and reproducibility comparable to rdRNA-seq (please refer to the revised version of Supplementary Fig. 4).

Moreover, to our knowledge, there is no scRNA-seq method using probes for rRNA depletion. Developing scRNA-seq methods using rRNA depletion is very difficult because loss of RNA in the RNA purification step is extremely severe. Indeed, the minimum input RNA amount for standard rRNA depletion RNA-seq kits is approximately 25 ng of total RNA, which is far greater than the typical RNA amount in single cells (1-10 pg). Thus, we chose to use NSRs to reduce rRNA reads (Armour et al., Nat Methods, 2009; Ozsolak et al., Genome Research, 2010). Therefore, we have added the following text to the Introduction section (p. 4, line 3):

In addition, how to reduce the sequence derived from ribosomal RNAs (rRNA) that accounts for most of the total RNA is a major task for establishing single-cell total RNA-seq. This issue is encountered because single-cell RNA-seq uses a trace amount of total RNA as a template, and it is difficult to apply general rRNA-depletion methods that cause loss of RNA.

Finally, we have changed the sentence “the RamDA-seq library still contains a relatively high proportion (10-25%) of rRNA sequences (Supplementary Fig. 3b).” in the Discussion section (p. 15, line 27) as follows:

the RamDA-seq library still contains a relatively high proportion (10-35%) of rRNA sequences (Supplementary Fig. 4b,c and Supplementary Fig. 12c)

Comment 2. Some more elaboration on the sample sizes would be desirable. How many diluted mESC RNA samples were generated with each protocol to calculate the CV² values in Figure 1E? I would expect around 100 samples at least, given that real plate-based scRNA-seq experiments would be expected to use the entire plate. A related concern is the number of plates/batches that were used, not just the number of wells on each plate. scRNA-seq data is subject to considerable plate effects (see <https://doi.org/10.1101/025528>, <https://doi.org/10.1038/srep39921>), and I would strongly recommend examining not just the well-to-well variability, but also the plate-to-plate variability, e.g., when plates are processed on different days, with different operators, etc.

Response: We agree that this point requires clarification. The number of diluted RNA samples was $n = 3$ or 4 for each scRNA-seq method. We showed the sample sizes in the revised version of Figure 1e.

In response to the concern raised by this reviewer, we also prepared ~96 diluted RNA samples (corresponding to the entire plate) with RamDA-seq, C1-RamDA-seq, and C1-SMART-seq v4 and calculated CV² values. RamDA-seq and C1-RamDA-seq showed high reproducibility (low CV² values) in most expression level bins (Supplementary Fig. 4f).

Moreover, in accordance with the reviewer's comment, we evaluated the plate-to-plate variability of RamDA-seq. Thus, we have added new experimental data to Supplementary Figure 16 in the revised manuscript. Two experimenters performed RamDA-seq with two plates of single-cell samples (consisting of ES and PrE cells) on two separate days (revised version of Supplementary Fig. 16a). Plate-to-plate variability was evaluated using the

correlation of expression levels and PCA. Although variabilities between plates were larger than variabilities within plates, plate-to-plate variability was much smaller than the variability between cell types (revised version of Supplementary Fig. 16d-f), and the proportion of variance explained in PCA was 0.6% (revised version of Supplementary Fig. 16g-h). We have added the following sentence to the Discussion section (p. 18, line 9):

We also evaluated plate-to-plate variability (batch effect) of RamDA-seq (Supplementary Fig. 16a). Although variabilities between plates were larger than variabilities within plates, plate-to-plate variability was much smaller than the variability between cell types, and the proportion of variance explained in principle component analysis (PCA) was 0.6% (Supplementary Fig. 16b-h).

Comment 3: With respect to the conclusion that RamDA-seq provides full-length coverage, Figure 2A may be misleading as the profile depends on the true length of transcripts. One can imagine that, even for a perfect technique that gave completely uniform coverage, the profile would not be flat if the underlying distribution of full transcript lengths was uneven. I would suggest an additional plot where the transcript lengths are scaled to a fixed interval. This ensures a protocol yielding full-length uniform coverage would also yield a flat profile.

Response: We apologize for confusing the reviewer by our insufficient explanation and misleading labeling. The profile in Figure 2a would be flat if a method gives complete uniform coverage because the y-axis represents the fraction of transcripts with mapped reads in $[x_i, x_{i+1}]$ bin in transcripts with $\geq x_i$ transcript length. To avoid confusion, we added Supplementary Figure 6, revised the legend of Figure 2a, and changed the y-axis label to “Percent of transcripts with mapped reads” (Fig. 2a). We believe that the transcript coverage against absolute transcript length (like Fig. 2a) provides essential and complementary information, especially for understanding cDNA synthesis bias dependent on transcript length. Thus, we kept the plot showing transcript coverage against absolute transcript length in the revised manuscript.

While reviewing our data analysis pipeline (as suggested by Reviewer #1’s Major comment 7), we found that the plot in the previous version of Figure 2a was created using RefSeq annotation. To ensure the consistency with the other parts of the manuscript, we replaced the previous plot with a plot created using GENCODE (revised version of Fig. 2a and Supplementary Fig. 4i). This revision did not change the conclusion.

On the other hand, we agree that additional information on the read coverage as the reviewer suggested would be valuable. Thus, we have added the transcript coverage of scRNA-seq methods against relative transcript positions to the revised version of Supplementary Figure. 7d. RamDA-seq showed uniform transcript coverage, similar to rdRNA-seq (revised version of Supplementary Fig. 7d).

Comment 4: The numbers on page 10 do not match up with the numbers in Figure 3B.

Response: The reviewer's comment is correct. We corrected the numbers in the revised manuscript (p. 10, line 6).

Comment 5: The linear regression to identify the sawtooth pattern for recursive splicing does not seem appropriate. In particular, I do not understand why the authors are using a dummy binary variable (page 52). It seems more logical to use the distance from the closest upstream RS site candidate as a real-valued covariate to model the coverage at each bin, given that coverage drops linearly with distance. A related issue is the claim that "the sawtooth wave pattern... weakened as differentiation progressed". While the decrease in coverage is clear, is this a specific reduction in recursive splicing or simply a general downregulation of the entire gene?

Response: We thank the reviewer for providing a suggestion on how to improve our analysis. In response to the first point in the reviewer's comment, we implemented the proposed model, which uses the distance from the closest upstream RS site candidate as a real-valued covariate and fitted the model against mapping data on *Robo2*, *Cadm1*, and *Magi1*. Additionally, we performed the Cox test for a non-nested model to evaluate the significance of fitting (briefly, the proposed model is selected against the baseline model if the difference in AIC (dAIC) is positive and $p < 1e-5$). Unfortunately, the proposed model failed to detect the sawtooth pattern in *Robo2* and *Magi1* (Letter Figure 1). The proposed model is forced to predict the same values at an RS site and its upstream annotated splicing junction; thus, it is likely to lack the flexibility to detect the sawtooth pattern of recursive splicing. Based on the new results, we would like to keep the previous version of the results.

Regarding the second point in this reviewer's comment and Reviewer #2's major comment 7, we reanalyzed the read coverages in detail. We found that the sawtooth wave pattern itself was robustly observed (revised version of Supplementary Fig. 13, middle panels) and that

the 'height' of the pattern was associated with the expression levels of host genes (revised version of Supplementary Fig. 13, left panels). Therefore, we have changed the sentence "Interestingly, the sawtooth wave pattern in the first intron of *Cadm1* weakened as differentiation progressed (Fig. 4b), whereas the pattern in the second intron of *Robo2* strengthened (Fig. 4c). The sawtooth patterns were also confirmed using averaged RamDA-seq read coverage for each time point (Fig. 4d-e). These results demonstrate that RamDA-seq can detect RS at the single-cell level and exemplify the potential of RamDA-seq to detect dynamics in RNA processing events." in the revised Results section (p. 11, line 27) as follows:

The averaged RamDA-seq read coverage for each time point showed that the sawtooth wave patterns persisted at all time points for *Cadm1* and *Magi1*, whereas the pattern was observed at only 48 and 72 h for *Robo2* (Fig. 4d,e, Supplementary Fig. 13). The height of the sawtooth wave pattern was associated with the expression level of host genes at all time points (Supplementary Fig. 13), while the pattern was not observed when gene expression was hardly detected (Supplementary Fig. 13). A simulation-based estimation of the sensitivity of RS detection showed that RS was robustly detected when host genes were sufficiently expressed (transcript per million (TPM) > 1), corroborating the above observations (Methods section; Supplementary Fig. 13, right panels).

Comment 6: For Figure 5B, I would like to see separate coverage profiles for the forward and reverse strands. This will confirm that they are indeed eRNAs, which are characterised by inverted bimodality around the enhancer. The "background" is also constructed from random genomic regions - how were these chosen?

Were these matched for GC content, mappability/sequenceability, DNA accessibility, etc.? Obviously, it would be trivial to obtain low "background" coverage if the background regions were defined in difficult-to-map regions such as telomeres.

Response: We agree that additional information on the coverage profiles for the forward and reverse strands as the reviewer suggested would be valuable. Regrettably, because RamDA-seq is a non-stranded sequencing method as mentioned in the Discussion section "It is also important to achieve strand-specific sequencing in RamDA-seq to distinguish overlapping transcripts." in the previous version of the manuscript, we are unable to perform the analysis. Thus, we have retained the original version of Figure 5b in the manuscript.

The background regions in the previous version of manuscript were selected randomly from intergenic regions, but we ignored GC content and other characteristics. We agree with the reviewer's concern that low-mappability regions would yield low read coverage. To eliminate this unwanted possibility, we prepared several other sets of background regions that have enhancer-like sequence characteristics and/or chromatin accessibility and made aggregation plots again. The read coverage in all background region sets was steadily and remarkably lower than that in ES-active enhancer regions (revised version of Supplementary Fig. 14c and 15c).

Comment 7: The read processing steps are very inconsistent across the different analyses. For some analyses, alignment was performed using HISAT2 (page 42), while for others, Bowtie2 was used (page 49). For quantification of gene expression, some analyses used Sailfish (pages 42, 45), others used Cufflinks (page 43) and others used featureCounts on aligned data (page 46). It would be advisable to standardize the quantification pipeline for all analyses, to avoid confusion and simplify the interpretation of the results. The use of Cufflinks seems particularly unnecessary, as I do not see many references to newly discovered unannotated transcripts in the manuscript.

Response: We thank this reviewer for raising several important points regarding data analysis pipelines. For quantification of transcript expression levels, we used Sailfish and Bowtie2+eXpress in different analyses in the previous version of manuscript, which might result in confusion in interpretability. As suggested the reviewer, we standardized the quantification pipeline to Sailfish and rewrote the method descriptions in the revised Methods section (“Quality assessment and general analysis of RamDA-seq and C1-RamDA-seq with cells” , p. 34, line 1; “Analyses of cell cycle data” p. 34, line 17; “Analyses of time-series data” p.35, line 17). Accordingly, we reanalyzed the cell cycle (revised version of Supplementary Fig. 8a-e) and time-series data (revised version of Figure 3a-e and Figure 5d-e). While the general conclusions did not change, we replaced the previous version of figures with revised version of the figures and made the corresponding changes in the revised manuscripts. For cell-cycle data, we have changed the sentence “This procedure yielded 4,919 oscillating transcripts, including 423 non-poly(A) transcripts (Supplementary Fig. 5b)” (Supplementary Note 5, p. 6, line 23) as follows:

This procedure yielded 6,736 oscillating transcripts, including 567 non-poly(A) transcripts (Supplementary Fig. 8c).

For time-series data, (1) we have changed the sentence “We identified 5,342 such transcripts, including 423 non-poly(A) transcripts (Fig. 3b), and divided the 5,342 transcripts into seven clusters based on expression patterns (Fig. 3c).” (p. 10, line 6) as follows:

We identified 7,580 such transcripts, including 458 non-poly(A) transcripts (Fig. 3b), and divided the 7,580 transcripts into seven clusters based on expression patterns (Fig. 3c).

(2) We have changed the sentence “*Neat1-001* was included in cluster 4 (Fig. 3c), and its expression level decreased specifically at 12 h in RamDA-seq and scRT-qPCR (Fig. 3e).” (p. 10, line 22) as follows:

The expression level of *Neat1-001* specifically decreased at 12 h in RamDA-seq and scRT-qPCR (Fig. 3e).

(3) The clusters in which *Pou5f1* and PrE maker genes were included were changed from clusters 5 and 3 to clusters 6 and 4, respectively. Thus, we have made the corresponding changes in the revised manuscript (Supplementary Note 7, p. 7, lines 26 and 31; Supplementary Note 8, p. 8, line 14). In addition, we have replaced *Col4a1* to *Col4a2* in the revised manuscript (Supplementary Note 8, p. 8, lines 11 and 13).

(4) The number of dynamically regulated eRNAs decreased (Fig. 5d) and, for interpretability, we set the number of clusters to five in the subsequent clustering (Fig. 5d,e). Accordingly, the clusters in which GATA4 motif was enriched were changed from 4 and 6 to 3 and 5, respectively. Thus, we have made the corresponding changes in the revised manuscript (p. 14, line 17) and have changed the sentence “We found 1,441 dynamically regulated eRNAs (FDR < 0.01) in seven clusters: three down-regulated clusters, three transiently up-regulated clusters, and one late up-regulated cluster (Fig. 5d-e).” (p. 14, line 11) and as follows:

We found 1,338 dynamically regulated eRNAs (FDR < 0.01) in five clusters: two down-regulated clusters, two transiently up-regulated clusters, and one late up-regulated cluster (Fig. 5d,e).

For analyses of recursive splicing and eRNA, we have established specialized pipelines based on the mapping data (by HISAT2) because, unlike expression level quantification, these analyses need read coverage data along genomic coordinates. To avoid confusion, we added sentences to the revised Methods section (p. 36, line 16; p. 38, line 13).

Cufflinks was applied to the bulk RNA-seq data (not single-cell RamDA-seq data) for transcriptome assembly. In the previous version of manuscript, the resulting gene models

were used to demonstrate the potential of RamDA-seq for detecting annotated and newly discovered non-poly(A) transcripts and to define a set of enhancer RNAs with enhancer-like histone modification. Thus, we disagree that the use of Cufflinks is unnecessary and would rather retain the previous version of the analysis. We hope that this issue does not cause any unnecessary trouble.

Comment 8: The selection of genes on page 48 is not coherent. Why do the authors identify the top HVGs within each phase, and then select the genes that are DE between phases? This will artificially encourage the separation of cells from different phases on the diffusion map. If the aim is to create an unbiased diffusion map, they should just use highly variable genes detected across the entire population of cells (from all phases). Otherwise, the authors should justify their choice of genes to construct a biased pseudotime.

Response: We appreciate the reviewer's concerns regarding this point. In response to the reviewer's comment, we conducted an 'unsupervised' diffusion map analysis using highly variable genes selected regardless of the derived cell-cycle phases. RamDA-seq could detect differences among cells in different cell-cycle phases (revised version of Supplementary Fig. 8a).

Our aim in the previous version of manuscript was to reconstruct a pseudotime specifically reflecting a subjective time along the cell cycle; thus, we chose the gene selection strategy in the previous version of manuscript. We would prefer to retain the previous version of manuscript. To clarify our aim and avoid confusion, we added descriptions of diffusion maps using both the 'unsupervised' and 'supervised' gene selection methods (revised version of Supplementary Fig. 8a,b), and we have changed the sentences "This analysis divided the cells according to their derived cell-cycle phases, although the cells in each phase showed variance (Supplementary Fig. 5a). Based on this observation, we hypothesized that the RamDA-seq data could be used to reconstruct a 'subjective time' for cells along the cell cycle. To address this possibility, we used DC1 as a proxy of subjective time for each cell along the cell cycle and searched for transcripts oscillating along DC1 by fitting a sine function (false discovery rate (FDR) < 0.01; Materials and Methods)." in the revised manuscript (Supplementary Note 5, p.6 line 12) as follows:

The cells were largely separated according to the derived cell-cycle phase (Supplementary Fig. 8a). This result indicates that RamDA-seq could detect variability among different cell-cycle phases within a single cell type.

Based on this observation, we hypothesized that the RamDA-seq data could be used to reconstruct a 'subjective time' for cells along the cell cycle. To address this possibility with use of information from a cell sorter, we first selected transcripts showing high variance among G1, S, and G2M, and performed diffusion map analysis. This procedure likely reconstructed a subjective time in each cell; the cells formed different clusters according to their derived cell, while the cells in each phase showed variance (Supplementary Fig. 8b). We then used DC1 as a proxy of subjective time for each cell along the cell cycle and searched for transcripts oscillating along DC1 by fitting a sine function (false discovery rate (FDR) < 0.01; see Methods).

We have noticed that the description of how to select highly variable genes in the previous version of manuscript was misleading. To avoid confusion, we have changed the sentence "For each transcript among the expressed genes, we first calculated the high CV of the averaged TPM within the G1, S, and G2M samples" in the revised manuscript (p. 34, line 21) as follows:

For each transcript among the expressed genes, we first calculated averaged TPM across cells in each cell cycle phase and then calculated the CV using the averaged TPM values for G1, S, and G2M.

Minor Comments

Comment 1: Genomic contamination is an important consideration for RNA-seq protocol development. There is no harm in showing the data as a Supplementary on the last line of page 6.

Response: In accordance with the reviewer's comment, we added quality control data for RamDA-seq library DNA, including the data from the RT enzyme minus and non-template control experiments in the revised manuscript (revised version of Supplementary Fig. 3). Contamination of genomic or environmental DNA did not occur.

Comment 2: Figure 1D uses a rather unconventional metric to examine the similarity between each scRNA-seq library and the bulk data (the number of genes with counts ≥ 10 and $|FC| \leq 2$). A more natural statistic would be the Pearson's correlation between the

counts, which can be directly interpreted as the proportion of variance explained. I suspect this would give similar results without the need for the arbitrary threshold on the fold-change. Otherwise, it would be necessary to show similar results at different thresholds.

Response: As suggested by the reviewer, we showed the Pearson correlation coefficients between each scRNA-seq method and bulk rdRNA-seq (revised version of Supplementary Fig. 4h). RamDA-seq and C1-RamDA-seq were most similar to bulk rdRNA-seq. We also showed the number of detected transcripts for each scRNA-seq method without any threshold for fold change (revised version of Supplementary Fig. 4d), which provided similar results to those in Figure 1d.

Comment 3: Supplementary Figure 5C is misleading as the expression values have simply been copy-and-pasted to form a continuum. I would suggest colouring the copies differently to avoid giving the idea that there is information on the cycle number for each cell.

Response: Thank you for your helpful suggestions for data visualization. To avoid confusion, we showed the data points once, colored the copies differently, changed the x-axis scale so that ticks corresponded to the boundary of cycles, and added arrows (revised version of Supplementary Fig. 5d,e).

Comment 4: Page 49 makes references to the cell cycle phases, while the section refers to time series data. I presume that this is a mistake.

Response: The reviewer's comment is correct. To correct for this mistake, we have changed the sentences "We conducted a diffusion map analysis as follows. First, we selected expressed transcripts with a TPM of at least 1 in at least 10% of the cells. For each transcript among the expressed genes, we first calculated the high CV of the averaged TPM within the G1, S, and G2M samples. For each cell cycle phase, we next selected the top 5000 high-CV transcripts with an averaged TPM that was higher than those for the other phases. For the selected transcripts, we performed diffusion map analysis using the 'destiny' R package. We used DC1 as the pseudotime, i.e., we treated the expression levels as a function of DC1." in the revised manuscript (p. 35, line 18) as follows:

We conducted a diffusion map analysis as follows: (1) we selected expressed transcripts with a TPM of at least 10 in at least 10% of cells. (2) We calculated the CV of TPM for each of the expressed transcripts and then selected the top 5,000 high-CV transcripts. (3) We performed diffusion map analysis on the expression data of the selected transcripts using the 'destiny' R package. (4) We used DC1 as the pseudotime, i.e., we treated the expression levels as a function of DC1.

Comment 5: Consider using TREAT (specifically, the glmTreat function) to test against a log-fold change threshold, rather than post-hoc filtering on the log-fold changes on page 44. See <https://support.bioconductor.org/p/62286/#62287> for a discussion of this topic by some of the edgeR authors.

Response: We thank the reviewer for providing a suggestion on how to improve our analysis. As suggested by the reviewer, we used the glmTreat function in the EdgeR package to identify non-poly(A) transcripts. We set the minimum log₂FC (a parameter of glmTreat) to 0.5 to ensure that the resulting non-poly(A) transcript set included positive controls (non-poly(A) transcripts confirmed by RT-qPCR in the revised version of Supplementary Fig. 9). Most of the non-poly(A) transcripts identified by glmTreat were included in the original non-poly(A) transcripts. Consistently, the conclusions did not change regardless of the testing/filtering methods, and we replaced the original figures with new figures (Fig. 2e, Fig. 3b,c, Supplementary Fig. 8c, Supplementary Fig. 10, Supplementary Fig. 14d,e, and Supplementary Fig. 15a) and replaced Supplementary Tables with new tables (Supplementary Data 8 and Supplementary Data 9).

RESPONSE TO REVIEWER 2:

We wish to express our appreciation for the reviewer's insightful comments, which have helped us to significantly improve the paper.

Major Comments

Comment 1: Important details of how the RT enzyme works are missing from this paper. While we understand that it is described in greater detail in the other paper (Hayashi et al., NAR, in review) by the same authors, it would be good to add a supplementary note

describing details about the underlying biology (e.g. how does strand displacement amplification work?)

Response: As suggested by the reviewer, to describe the details of the RT-RamDA reaction, we added a section “The principle of RT with random displacement amplification (RT-RamDA)” to the revised Supplementary Note 1.

Comment 2: The authors claim that “RamDA-seq protocol did not produce a library of DNA derived from genomic DNA or environmental DNAs (data not shown)”. Can the authors provide some evidence of this claim? Supplementary Fig 3c shows that the proportion of intronic/intergenic reads is higher in RamDA-seq (>40%) than other methods. How do we reconcile these two statements? Some commentary is essential.

Response: We confirmed that the RT enzyme minus and non-template control experiment did not produce library DNA (revised version of Supplementary Fig. 3), indicating that contamination of genomic or environmental DNA did not occur. Therefore, the QC metrics of scRNA-seq methods (Fig. 1 and revised version of Supplementary Fig. 4) indicate that the observed high proportion of intronic and intergenic reads, which is also a hallmark of rdRNA-seq, reflect the resemblance of RamDA-seq to rdRNA-seq. We believe that these reads were derived from transcripts, not endogenous genomic DNA.

In accordance with Reviewer #1’s major comment 7, we changed the data set of transcript annotation from Refseq to GENCODE to ensure consistency through the manuscript. Thus, we have replaced the previous version of Supplementary Figure 3c with the revised version of Supplementary Figure 4i. However, none of our conclusions changed. Finally, we have changed the sentence “Read distributions, especially proportions of intronic and 3’ UTR regions” in the Results section (p. 7, line 20) as follows:

Read distributions, especially high proportions of intronic, 5’ UTR and intergenic regions, showed that RamDA-seq resembled rdRNA-seq

Comment 3: For Fig. 1d and 1e, I believe a useful (and easy) control would be to perform RamDA-seq on 1 microgram RNA and compare its performance to the gold standards (rdRNA-seq and paRNA-seq)

Response: As suggested by the reviewer, we compared RamDA-seq and rdRNA-seq prepared with equal amounts of input RNA. Since RT-RamDA and a Nextera XT library preparation kit have limitations regarding the input amount of template, it is not easy to compare its performance with standard RNA-seq using the same amount of input RNA. Therefore, considering the maximum amount of input RNA, we prepared RamDA-seq libraries using 1 ng of RNA and compared their performance with that of rdRNA-seq libraries using 1 or 10 ng of RNA. Compared with rdRNA-seq (1 ng), RamDA-seq (1 ng) showed a higher proportion of rRNA reads (revised version of Supplementary Fig. 5 a), which was the case when comparing RamDA-seq (10 pg) with rdRNA-seq (1 µg) (revised version of Supplementary Fig. 4b,c and revised version of Supplementary Fig. 12c). Despite the difference, RamDA-seq (1 ng) and rdRNA-seq (1 ng) showed comparable sensitivity (number of detected transcripts) and reproducibility (SCC, PCC and CV²) (revised version of Supplementary Fig. 5b-d). Moreover, the correlation between RamDA-seq (1 ng) and rdRNA-seq (1 ng or 10 ng) was as high as that between RamDA-seq (10 pg) and rdRNA-seq (1 µg) (revised version of Supplementary Fig. 4h and revised version of Supplementary Fig. 5c). These results support our previous conclusion that RamDA-seq is a single-cell total RNA-seq method that resembles rdRNA-seq. We have added the following text to the Results section (p. 7, line 23):

We also compared large-volume-inputted RamDA-seq with rdRNA-seq using 1 ng of RNA. When we compared RamDA-seq and rdRNA-seq with the same input amount, there was hardly any difference in sensitivity and reproducibility except for the contamination rate of rRNA (Supplementary Fig. 5).

Comment 4: It is not clear if the data shown on Fig. 2b is an average across multiple single cells for RamDA-seq or coverage within a single cell. It would be useful to see what the coverage within example single cells looks like. Also it is not clear why Mdb1/Mdn1 (the name of the gene is mismatched between the figure and the text) was chosen? Can the authors comment on its significance?

Response: We completely agree and apologize for this omission. We originally showed the read coverage in 10 pg data (n= 1) (which corresponds to a single cell) (Fig. 2b). To clarify this point, we added the numbers of samples of each scRNA-seq (n= 1) to the revised version of Figure 2b.

We first selected 25 expressed long transcripts as transcripts whose TPM ≥ 5 in rdRNA-seq in mESC and whose transcript length ≥ 10 kb. We further selected *Mdn1*, which has the highest number of exons (102 exons) among the expressed long transcripts, for visualization. In response to Reviewer #2's minor comment 4, we showed the read coverage of 25 other expressed long transcripts (revised version of Supplementary Data 2 and 3 for 10 pg RNA and mESC data, respectively). We also added text to the legend of Figure 2b in the revised manuscript as follows:

We selected *Mdn1* as the gene with the highest number of exons (102 exons) in the 25 genes with length ≥ 10 kb and TPM ≥ 5 in rdRNA-seq results.

Comment 5 Are non-poly(A) transcripts enriched specifically in Clusters 1,2 and 5 in Fig. 3d. If so, why?

Response: In response to Reviewer #1's major comment 7, we changed the expression quantification method and reanalyzed the data. Although the proportion of non-poly(A) transcripts in each cluster changed, non-poly(A) transcripts were included in all clusters irrespective of the quantification methods (revised version of Fig. 3c). This result suggests that non-poly(A) transcripts might contribute to various cell functions. Accordingly, we have changed the sentence "Closer inspection of clusters 1, 2, and 5 revealed that these three clusters contained many non-poly(A) transcripts, as exemplified by two unannotated intergenic non-poly(A) transcripts and *Hist1h1a* (Fig. 3d)." in the revised manuscript (p. 10, line 9) as follows:

The dynamically regulated non-poly(A) transcripts were spread in all clusters with various expression patterns, suggesting that non-poly(A) transcripts are involved in various cell functions. Using single-cell RT-qPCR, we validated the observed expression changes in several of the non-poly(A) transcripts, including two unannotated intergenic non-poly(A) transcripts (clusters 1 and 2) and *Hist1h1a* (cluster 5) (Fig. 3d and Supplementary Fig. 12g).

Comment 6: Related to Fig. 3, a discussion on the biological significance of Neat1 is missing. The authors should relate to what was published

Response: In accordance with the reviewer's comment, we added a new paragraph (p. 17, line 1) to the revised version of the Discussion as follows:

Neat1 is an architectural component of paraspeckle nuclear bodies³⁸, which regulate gene expression via capture of A-to-I edited mRNAs³⁹ and transcription factors⁴⁰, and is required for corpus luteum formation and establishment of pregnancy in mice⁴¹. The long non-poly(A) isoform *Neat1-001*, not the short poly(A) isoform *Neat1-002*, is essential for the formation of paraspeckles⁴². Although the two isoforms are transcribed from the same promoter, they show different expression patterns, and *Neat1-001* is expressed only in a small subpopulation of cells in adult mouse tissues²⁶. Therefore, distinguishing the expression of the two isoforms of *Neat1* at the single-cell level is critical for studying their functions. In this study, RamDA-seq's sensitivity to non-poly(A) transcripts and full-length transcript coverage distinguished the expression of the two isoforms and revealed dynamic and differential regulation of the long non-poly(A) isoform (Fig. 3). These results suggest that RamDA-seq could be beneficial for investigation of temporal and spatial expression patterns of long non-poly(A) RNAs in single cells.

Comment 7: The authors claim that “the sawtooth wave pattern in the first intron of Cadm1 weakened as differentiation progressed, whereas the pattern in the second intron of Robo2 strengthened”. But this could merely be the result of changing expression levels. In other words, the data as presented are consistent with the fact that the recursive splicing stays exactly the same, it's only that it cannot be detected when global expression levels are low. Can the authors comment on the sensitivity of RS detection as a function of expression levels?

Response: In response to the reviewer's comment and Reviewer #1's major comment 5, we reanalyzed the read coverages in detail. The sawtooth wave pattern itself was robustly observed (revised version of Supplementary Fig. 13, middle panels), and the 'height' of the pattern was associated with host gene expression levels (revised version of Supplementary Fig. 13, left panels).

To estimate the sensitivity of RS detection as a function of host gene expression levels, we then performed a simulation in which intronic read coverage data were repeatedly subsampled (100 times per subsampling fraction), and the success rate of RS detection was calculated (Methods section). Then, the aggregated expression levels of host genes at each time point were mapped to the estimated sensitivity curve (revised version of Supplementary

Fig. 13, right panel). Note that the fraction of intronic reads was converted to host gene expression levels (TPM). The result shows that the sensitivity of RS detection is a sigmoid-like monotonically increasing function of host gene expression (revised version of Supplementary Fig. 13, right panel) and coincides with RS detection at each time point for *Robo2*, *Cadm1*, and *Magi1* (revised version of Supplementary Fig. 13, middle panel). The detection limit varied from gene to gene. Altogether, we were able to estimate the sensitivity of RS detection. Based on these results, we have changed the sentences “Interestingly, the sawtooth wave pattern in the first intron of *Cadm1* weakened as differentiation progressed (Fig. 4b), whereas the pattern in the second intron of *Robo2* strengthened (Fig. 4c). The sawtooth patterns were also confirmed using averaged RamDA-seq read coverage for each time point (Fig. 4d-e). These results demonstrate that RamDA-seq can detect RS at the single-cell level and exemplify the potential of RamDA-seq to detect dynamics in RNA processing events.” in the revised manuscript (page 11, line 27) as follows:

The averaged RamDA-seq read coverage for each time point showed that the sawtooth wave patterns persisted at all time points for *Cadm1* and *Magi1*, whereas the pattern was observed at only 48 and 72 h for *Robo2* (Fig. 4d,e, Supplementary Fig. 13). The height of the sawtooth wave pattern was associated with the expression level of host genes at all time points (Supplementary Fig. 13), while the pattern was not observed when gene expression was hardly detected (Supplementary Fig. 13). A simulation-based estimation of the sensitivity of RS detection showed that RS was robustly detected when host genes were sufficiently expressed (transcript per million (TPM) > 1), corroborating the above observations (Methods section; Supplementary Fig. 13, right panels).

Comment 8: Also, can the authors detect heterogeneity across single cells in the presence/absence of RS within a specific gene?

Response: We appreciate the reviewer's comment regarding this point. In response to the reviewer's comment, we performed RS detection analysis using single-cell data after filtering cells with a small number of intronic reads (based on RS detection sensitivity). RS was detected in a subpopulation of cells (revised version of Supplementary Data 5). This result indicates that RamDA-seq can detect RS even in individual single cells when a robust sawtooth pattern is observed. Moreover, we observed cell-to-cell heterogeneity in read coverage patterns around RS sites, suggesting that some cells showed recursive splicing

and other cells showed normal splicing (Supplementary Data 5). These results indicate that RamDA-seq can detect cell-to-cell heterogeneity in RS.

To report these results, we have added a new paragraph to the Results (p. 12, line 9) section as follows:

Next, we attempted to address whether RamDA-seq can detect RS even in each single cell. Based on the RS detection sensitivity estimated above, we selected cells with sufficient intronic reads (RS detection probability > 0.95) (Supplementary Data 5, Methods section). We fitted linear regression models against the RamDA-seq read coverages of each single cell in *Cadm1*, *Robo2*, and *Magi1*. RS was detected in a subpopulation of cells (71 of 149 cells for *Cadm1*, 12 of 54 cells for *Magi1*, 1 of 1 cell for *Robo2*) although many cells in which RS was not detected also appeared to show the sawtooth pattern. However, interestingly, some other cells showed monotonically decreasing patterns, which correspond to 'normal' splicing (Fig. 4a). The monotonically decreasing pattern was also observed even when we filtered cells with a more stringent threshold of intronic reads (Supplementary Data 5). These observations raise the possibility of cell-to-cell heterogeneity in RS. Therefore, further investigation is needed to reveal the mechanisms and significance of the observed heterogeneity in RS. Collectively, these results demonstrate that RamDA-seq can detect RS in single cells.

We have also added a new paragraph to the Discussion section (p. 17, line 15) in the revised manuscript as follows:

Unexpectedly, we observed cell-to-cell heterogeneity in read coverage patterns around RS sites, suggesting that some cells showed recursive splicing, and other cells showed normal splicing (Supplementary Data 5). These results indicate that RamDA-seq can detect cell-to-cell heterogeneity in RS and could help to address the mechanisms and relationship between transcription and splicing. Toward these goals, several important challenges remain. Given that some cells in which RS was not detected showed weak sawtooth patterns, RamDA-seq highlights the limitation of the current linear regression model used to detect RS in this study and the need for further improvement in computational methods to robustly detect RS using single-cell data. Another challenge is to experimentally and computationally distinguish biological and technical variabilities in RS at the single-cell level. We will address these challenges in the future.

Comment 9: Lastly, but importantly, given the high proportion of rRNA, intronic and intergenic reads in RamDA-seq, can the authors comment on the depth per cell needed to get useful information out?

Response: We evaluated the number of detected transcripts against the number of reads for scRNA-seq methods (10 pg) by subsampling reads in FASTQ files. RamDA-seq outperforms the other scRNA-seq methods with >1 M reads per cell (revised version of Supplementary Fig. 4e). Moreover, with ~4 M reads per cell (cell-cycle and time-series data), RamDA-seq, as single-cell total RNA-seq, yield reads from non-poly(A) transcripts, introns and intergenic regions and provides information on recursive splicing and enhancer RNA. We note that 4 M reads per cell is typical for plate-based single-cell RNA-seq (For example, 96 cells in 1 run on NextSeq yields approximately 4 M reads per cells). As suggested by the reviewer, to explain the depth per cell needed to obtain useful information, we have added a paragraph to the Discussion section in the revised manuscript (page 17, line 7) as follows:

Based on a subsampling simulation, with > 1 M reads per cell, RamDA-seq detects more transcripts than the other scRNA-seq methods (Supplementary Fig. 4e). Moreover, with ~4 M reads per cell, RamDA-seq yields reads from non-poly(A) transcripts, introns and intergenic regions and provides beneficial information regarding unannotated intergenic transcripts, RS and enhancer RNA (Fig. 3-5). Given that ~4 M reads per cell are typical for plate-based single-cell RNA-seq (For example, 96 cells in 1 run on NextSeq yields approximately 4 M reads per cell), these results demonstrate that RamDA-seq needs just normal sequencing runs to provide useful information regarding gene expression, transcriptional regulation, and RNA processing.

Minor Comments

Comment 1: The often used phrase “RamDA-seq is the single-cell RNA-seq method” very strange. This does not convey any meaning and I am sure is an error. Stated this way, it appears that there is no other single-cell RNA-seq method other than RamDA-seq. I would suggest a rephrasing as follows: “Altogether, based on the substantial resemblance, we conclude that RamDA-seq is a robust single-cell total RNA-seq method”

Response: In accordance with the reviewer's comment, we changed the description throughout the revised manuscript.

Comment 2: For comprehensiveness, it would be good if Fig. 2b involved SUPER-seq. Also can the authors share a few more representative examples like Fig. 2b (which is impressive) in the supplement?

Response: In accordance with the reviewer's comment, we added SUPeR-seq to the comparison in the revised version of Figure 2b. We also showed other examples in the revised version of Supplementary Data 2 and 3 (for 10 pg RNA and mESC data, respectively).

Comment 3: The wording of Fig. 2c within the text is not clear. What does "longer range of transcripts" mean? I guess the authors are referring to a longer range of transcripts binned by length – clarification would help.

Response: In accordance with the reviewer's comment, we have changed the sentence "In addition, the fraction of exonic regions covered by the reads also indicated that RamDA-seq covered a wider range of transcripts than did the other methods (Fig. 2c and Supplementary Fig. 4)." (p. 8, line 20) as follows:

In addition, the fraction of exonic regions covered by the reads indicated that RamDA-seq covered a higher fraction of exonic regions than did the other methods in all length bins (Fig. 2c, Supplementary Fig. 7a-c).

Original model

Proposed model

Robo2

Cadm1

Magi1

REVIEWERS' COMMENTS:

Reviewer #1 (Remarks to the Author):

I am satisfied with the authors' revisions to the manuscript, which have largely addressed my concerns. I only have a few minor comments:

Supplementary Figure 1: "Quarity control" is misspelt.

Supplementary Figure 4: What is the difference between % of rRNA reads in FASTQ and % of mapped reads in rRNA sequences? This was not apparent from reading the caption or methods. I would have expected the latter to be larger than the former, given that rRNA reads must map (otherwise they would not be identified as mapped reads) but other reads may or may not map.

Supplementary Figure 16: This looks good, but I'm not clear on how the authors concluded that "variabilities between plates were larger than variabilities within plates". The within-plate correlations in D and F look very similar to the between-plate correlations, which suggests - encouragingly - that the plate effects are modest at worst.

Reviewer #2 (Remarks to the Author):

I have reviewed the author's responses carefully and am satisfied. I recommend this paper for publication.

RESPONSE TO REVIEWER 1:

Original reviewer comments appear below in italics with responses in blue.

We wish to express our appreciation for the reviewer's comments, which have helped us to significantly improve the paper.

Supplementary Figure 1: "Quarity control" is misspelt.

Response: We corrected the spelling in Supplementary Figure 1.

Supplementary Figure 4: What is the difference between % of rRNA reads in FASTQ and % of mapped reads in rRNA sequences? This was not apparent from reading the caption or methods. I would have expected the latter to be larger than the former, given that rRNA reads must map (otherwise they would not be identified as mapped reads) but other reads may or may not map.

Response: We apologized for the misleading description in the legend text of Supplementary Figure 4c. In Supplementary Figure 4c, only reads uniquely mapped to genome were used. Specifically, the values in y-axis is calculated by dividing the number of uniquely mapped reads overlapping the rRNA gene annotation by the number of uniquely mapped reads. Because rRNA sequences tend to be multi-mapped, it is expected that the percent in Supplementary Figure 4c to be smaller than that in Supplementary Figure 4b.

To avoid confusion, we revised the y-axis label in Supplementary Figure 4c and the corresponding legend text as follows:

(c) The percentage of uniquely mapped reads that were overlapped with rRNA gene annotations on genome.

Supplementary Figure 16: This looks good, but I'm not clear on how the authors concluded that "variabilities between plates were larger than variabilities within plates". The within-plate correlations in D and F look very similar to the between-plate correlations, which suggests - encouragingly - that the plate effects are modest at worst.

Response: We thank the reviewer for the encouraging comment. We changed the sentences "Although variabilities between plates were larger than variabilities within plates, plate-to-plate variability was much smaller than the variability between cell types, and the proportion

of variance explained in principle component analysis (PCA) was 0.6% (Supplementary Fig. 16b-h)." in the Discussion section as follows:

The variabilities between plates were similar to the variabilities within plates, suggesting that the plate effects are modest at worst. Plate-to-plate variability was much smaller than the variability between cell types, and the proportion of variance explained in principal component analysis (PCA) was 0.6% (Supplementary Fig. 16b-h).